# Synthesis of Novel Tritopic Hydrazone Ligands: Spectroscopy, Biological Activity, DFT, and Molecular Docking Studies

**DOI:** 10.3390/molecules27051656

**Published:** 2022-03-02

**Authors:** Sharmin Akther Rupa, Md. Rassel Moni, Md. Abdul Majed Patwary, Md. Mayez Mahmud, Md. Aminul Haque, Jamal Uddin, S. M. Tareque Abedin

**Affiliations:** 1Department of Chemistry, Comilla University, Cumilla 3506, Bangladesh; sharminrupa@cou.ac.bd (S.A.R.); monirassel@gmail.com (M.R.M.); 2Faculty of Pharmaceutical Science, Tokushima University, Tokushima Shi 770-0026, Japan; hemmayez@yahoo.com; 3Department of Chemistry, Jagannath University, Dhaka 1100, Bangladesh; amin2k12@chem.jnu.ac.bd; 4Department of Natural Sciences, Center for Nanotechnology, Coppin State University, Baltimore, MD 21216, USA; 5Department of Chemistry, Jahangirnagar University, Savar 1342, Bangladesh

**Keywords:** DFT, vibrational frequencies, FMO, tritopic, polydentate, molecular docking

## Abstract

Polytopic organic ligands with hydrazone moiety are at the forefront of new drug research among many others due to their unique and versatile functionality and ease of strategic ligand design. Quantum chemical calculations of these polyfunctional ligands can be carried out in silico to determine the thermodynamic parameters. In this study two new tritopic dihydrazide ligands, N’2, N’6-bis[(1*E*)-1-(thiophen-2-yl) ethylidene] pyridine-2,6-dicarbohydrazide (L1) and N’2, N’6-bis[(1*E*)-1-(1H-pyrrol-2-yl) ethylidene] pyridine-2,6-dicarbohydrazide (L2) were successfully prepared by the condensation reaction of pyridine-2,6-dicarboxylic hydrazide with 2-acetylthiophene and 2-acetylpyrrole. The FT-IR, ^1^H, and ^13^C NMR, as well as mass spectra of both L1 and L2, were recorded and analyzed. Quantum chemical calculations were performed at the DFT/B3LYP/cc-pvdz/6-311G+(d,p) level of theory to study the molecular geometry, vibrational frequencies, and thermodynamic properties including changes of *∆H*, *∆S,* and *∆G* for both the ligands. The optimized vibrational frequency and (^1^H and ^13^C) NMR obtained by B3LYP/cc-pvdz/6-311G+(d,p) showed good agreement with experimental FT-IR and NMR data. Frontier molecular orbital (FMO) calculations were also conducted to find the HOMO, LUMO, and HOMO–LUMO gaps of the two synthesized compounds. To investigate the biological activities of the ligands, L1 and L2 were tested using in vitro bioassays against some Gram-negative and Gram-positive bacteria and fungus strains. In addition, molecular docking was used to study the molecular behavior of L1 and L2 against tyrosinase from *Bacillus megaterium.* The outcomes revealed that both L1 and L2 can suppress microbial growth of bacteria and fungi with variable potency. The antibacterial activity results demonstrated the compound L2 to be potentially effective against *Bacillus* *megaterium* with inhibition zones of 12 mm while the molecular docking study showed the binding energies for L1 and L2 to be −7.7 and −8.8 kcal mol^−1^, respectively, with tyrosinase from *Bacillus megaterium.*

## 1. Introduction

Polytopic ligands containing hydrazide-hydrazone moiety (-CO-NHN=CH-) are important for new drug development [1,2,3,4,5,6] because their polyfunctional nature offers multifarious synthetic ways to derivatize such organic molecules towards suitable and effective drug–receptor interaction. The derivatives of hydrazide-hydrazone moiety with heterocyclic systems possess a range of biological activities; namely, anti-microbial, anti-mycobacterial, antitubercular, anticonvulsant, anticholinesterase [1], antiplatelet, and more importantly antitumor [5,7]. Transition metal complexes derived from such types of ligands have been widely studied since they also demonstrate significant biological and pharmacological properties [8,9,10,11]. Self-assembly of transition metals with multifunctional polydentate ligands has resulted in a successful paradigm of single-step synthesis of a new class of clusters of well-defined aesthetical architectures bearing spectacular electronic, catalytic, photophysical, photochemical, and magnetic properties [10,11,12]. In addition, depending on the relative orientation of the terminal donor groupings, exotic helicate structures, self-assembled clusters, and grids can be synthesized [10,11,12,13,14,15,16,17,18,19]. Nonetheless, the foremost synthetic challenge in this field is to design and synthesize new polytopic ligands with appropriate donor groupings placed strategically in place for metal coordination. In general, pyrrole, thiophene, and their organic and metal cluster derivatives are recognized to present a wide range of biological activities [20,21,22,23,24,25,26,27,28,29,30,31,32,33]. Therefore, the design and synthesis of novel ligands containing the thiophene and pyrrole rings have attracted great interest.

In recent years, computational chemistry along with sophisticated computational methods have been widely used in solving and modeling chemical reactivity, establishing structure-reactivity correlations, and in predicting vibrational, electronic, and thermodynamic properties of the compounds in the biological and chemical environments [34]. The methods are also used as secondary evidence to support analytical and experimental data [35]. Density functional theory (DFT) has been extensively employed for the calculation of various molecular properties such as molecular structure, UV-Vis, NMR, FT-IR, and Raman activities of biological compounds. Often such responses are measured using harmonic force fields. These methods provide relatively accurate molecular structures and energies using a suitable basis set compared to the conventional ab initio restricted Hartree–Fock (RHF) and Moller–Plesset second-order perturbation theory (MP2) calculations. DFT predicts relatively accurate vibrational wavenumbers for polyatomic molecules with moderate computational effort [36,37,38].

In the present study, the design and synthesis of two novel tritopic hydrazone derivatives, L1 and L2, are described with spectroscopic, DFT, and molecular docking studies. The structural characterization of L1 and L2 were experimentally accomplished by FT-IR, (^1^H and ^13^C) NMR, and mass spectroscopic techniques. Quantum mechanical investigations were performed in detail by computational methods exploring the optimized molecular geometries, molecular electrostatic potential (MEP) mapping with atomic charges, IR frequencies, NMR chemical shifts (^1^H and ^13^C), HOMO-LUMO energies, and thermodynamic parameters. Dipole moments of both L1 and L2 were also investigated using the DFT method at the B3LYP level with the 6-311G+(d,p) basis set. The experimental data obtained from FT-IR and NMR chemical shifts (^1^H and ^13^C) were compared to the calculated results found from the DFT method. In addition to that, the compounds were tested in vitro against some Gram-negative and Gram-positive bacteria and two fungi. A molecular docking methodology was used to study the molecular behavior of L1 and L2 against tyrosinase from *Bacillus*
*megaterium*.

## 2. Experimental Data

All chemicals were purchased from Sigma-Aldrich and used as received without further purification. Infrared (IR) spectra were recorded on a Shimadzu (FTIR) Prestige-21 spectrophotometer (Kyoto, Japan) (range: 4000–400/cm, using KBr disk); ^1^H and ^13^C-NMR spectra were recorded on a Bruker DPX-400 spectrophotometer (Bermen, Germany) using tetramethyl silane as an internal reference. NMR spectra were recorded on DMSO solvent. Mass spectra were obtained from VG Micro mass 7070HS (EI) and HP1100MSD (LCMS) spectrometers (Santa Clara, CA, USA).

### 2.1. Synthesis of Ligands

#### 2.1.1. Synthesis of Ligand (L1)

The synthesis of pyridine-2,6-dicarboxylic dihydrazide was carried out according to the synthetic procedure reported and published by Thompson et al. [39]. A solution of pyridine-2,6-dicarboxylic hydrazide (590.5 mg, 3.03 mmol) in methanol (30 mL) was added to a solution of 2-acetylthiophene (834.0 mg, 7.64 mmol) in methanol (10 mL) with continuous stirring. The resulting mixtures were stirred at 60 °C over a period of 18 h (Figure 1). A yellow precipitate was formed during the reaction, which was collected by filtration and dried under vacuum. (Yield: 92%). IR (KBr disk, cm^−1^): 3327 (ν NH), 3076 (ν C–H), 1682 (ν C=O), 990 (ν thiophene). ^1^H-NMR (400 MHz, DMSO-d_6_, ppm): δ 11.43 (s, 2H), 8.34 (m, 2H), 8.28 (m, 1H), 7.65 (d, 2H, *J* = 4.8 Hz), 7.61 (d, 2H, *J* = 3.2 Hz), 7.14 (t, 2H, *J* = 8.8 Hz), 2.51 (s, 6H). ^13^C-NMR (100 MHz, DMSO-d_6_, ppm): 159.44 (C=O), 154.55 (Ar-C), 148.91 (Ar-C), 143.22 (–C=N–), 140.48 (Ar-C), 130.06 (Ar-C), 129.44 (Ar-C), 128.20 (Ar-C), 125.75 (Ar-C), 15.32 (–CH_3_). Mass spectrum (*m*/*z*): 434.07 (MNa^+^), 305.41, 217.37.

#### 2.1.2. Synthesis of Ligand (L2)

L2 was prepared by refluxing the mixtures of a solution containing pyridine-2,6-dicarboxylic dihydrazide (390.33 mg, 2.0 mmol) in methanol (40 mL) and a solution containing 2-acetylpyrrole (437.43 mg, 4.0 mmol) in methanol (10 mL). The reaction mixture was stirred for 18 h under reflux and was then cooled to room temperature. A dark yellow precipitate was formed during the reaction, which was collected after 48 h by filtration and dried under vacuum. (Yield: 89%). IR (KBr disk, cm^−1^): 3501 (ν NH in pyrrole), 3456 (ν_sym_ NH), 3416 (ν_asym_ NH), 3099 (ν C–H), 1678 (ν C=O), 835 (ν pyrrole). ^1^H-NMR (400 MHz, DMSO-d_6_, ppm): δ 11.41 (s, 2H), 11.34 (s, 2H), 8.35 (m, 2H), 8.27 (m, 1H), 6.93 (m, 2H), 6.67 (m, 2H), 6.15 (m, 2H), 2.39 (s, 6H). ^13^C-NMR (100 MHz, DMSO-d_6_, ppm): 159.11 (C=O), 153.65 (Ar-C), 149.25 (–C=N–), 140.31 (Ar-C), 130.04 (Ar-C), 125.26 (Ar-C), 123.05 (Ar-C), 112.83 (Ar-C), 109.34 (Ar-C), 14.82 (–CH_3_). Mass spectrum (*m*/*z*): 400.15 (MNa^+^), 305.36, 217.38.

#### 2.1.3. Antimicrobial Activity Assay

In vitro antimicrobial activity of the synthesized ligands was evaluated by the agar disc diffusion method [40]. Mueller–Hinton agar (MHA) medium (HIMEDIA, India) was used as a control medium for testing against bacteria and potato dextrose agar (PDA) medium (HIMEDIA, India) was used for fungal strains. After preparation, the MHA and PDA media were incubated for 24 h and contaminations were checked. After incubation, the test organism was inoculated using sterile cotton bars on media. The sample discs were put gently on pre-inoculated agar plates and aerobically incubated for 24 h at 37 °C for the antibacterial and for 48 h at 26 °C for the antifungal assay. Dimethyl sulfoxide (DMSO) was used as the control. Each disc was loaded with 25 µL of sample solution in DMSO containing 300 µg of synthesized compounds. An amount of 10 µL of ceftriaxone and amphotericin-B solutions containing 50 µg each in DMSO were loaded per disc for antibacterial and antifungal assays as the positive control, respectively. The diameter of the inhibition zones in mm circling the disc were measured. Two Gram-positive *Staphylococcus aureus* (cars-2) and *Bacillus*
*megaterium (BTCC-18)*, two Gram-negative *Escherichia coli* (carsgn-2) and *Salmonella Typhi (K-323130)* bacteria, and two fungal strains *Trichoderma harzianum* (carsm-2) and *Aspergillus niger* (carsm-3) were used in this study. 

## 3. Computational Details

### 3.1. Geometry Optimization 

Optimized structures for L1 and L2, calculated at the B3LYP/6-311G+(d,p) level of theory in the gas phase, are presented in Figure 1. All calculations were conducted with the Gaussian 09 software package [41]. The complete geometry optimization and subsequent vibrational frequency calculations were performed using DFT employing Becke’s (B) [42] exchange functional combining Lee, Yang, and Parr’s (LYP) correlation functional [43] with standard cc-pvdz/6-311G+(d,p) basis sets. The absence of imaginary frequencies confirmed that the stationary points correspond to minima on the potential energy surface. All the optimized geometry corresponding to medium on the potential energy surface were obtained by solving self-consistent field equations iteratively. 

### 3.2. Protein-Ligand Docking

#### 3.2.1. Ligand and Protein Preparation

The structures of L1 and L2 were fully optimized using Gaussian 09 software at the B3LYP/6-311G+(d,p) level. The 3D crystal structure of tyrosinase from *Bacillus*
*megaterium* (PDB ID: 4j6u; resolution: 2.5Å, Chain A, B) was obtained in pdb format from the online RCSB protein data bank (PDB) database. The structure was verified, and an energy minimization was performed with the Swiss-Pdb Viewer software packages (version 4.1.0) [44], since the crystal structure contains a variety of issues related to improper bond order, side chain geometry, and missing hydrogen atoms. Prior to docking, all the heteroatoms and water molecules were removed from the crystal structure using PyMol (version 1.3) software packages [45]. The active binding pocket of tyrosinase was predicted by CASTp—the highest pocket area and volume are 95.432 Å^2^ and 137.877 Å^3^, respectively [46]. The binding site residues predicted by CASTp for tyrosinase were used for grid generation. Both the structures of the proteins and ligands were saved in .pdbqt format by AutoDock Vina (version 1.1.2, 11 May 2011) for docking analysis [47]. 

#### 3.2.2. Molecular Docking Analysis

The docking calculations were performed using default parameters and 8 docked conformations were generated for both compounds. The energy calculations were performed by genetic algorithms. Nonpolar hydrogen atoms, Gasteiger partial charges, rotatable bonds, and grid boxes with dimensions 66.57 × 58.25 × 84.98 Å^3^ were created on the tyrosinase with the support of Auto Dock Tools 1.1.2 and spacing of 0.3750 Å. The docked conformation of the respective protein conformer with the lowest binding free energy and root-mean-square deviation value (RMSD) 0.0 Å was analyzed using PyMOL Molecular Graphics System (version 1.7.4) and Accelrys Discovery Studio 4.1 [48].

## 4. Results and Discussion

### 4.1. Optimized Geometries

The selected bond distances (Å) and bond angles (°) of compounds L1 and L2 are presented in the Appendix A. Significant changes in bond distances were observed for specific reaction sites of the products compared to the reactants. Both the compounds were planar as expected, which was evident from the dihedral angles of the optimized structures showing that there is no twisting between the benzene ring and the substituent groups. Furthermore, the structures of the compounds revealed the preferential existence of both keto-enol forms. According to the optimization result, the short C–O bond distances indicated the retention of their keto form, and the CO groups were co-planar with the benzene rings [49]. For instance, the calculated C(10)-O(12) bond distances were 1.21 and 1.22 Å in L1 and L2, respectively, which were compared to the C(10)-O(11) bond distances (1.26 Å) in the reactant, pyridine-2,6-dicarboxylic hydrazide (see Appendix A). The calculated C–O bond lengths of reactants and products are in good agreement with the earlier reported length of 1.23 Å [50,51]. A similar pattern was observed for N–N bond lengths (N(14)-N(16) (~1.36 Å)) in both L1 and L2 [50,52,53]. The N-N bond length (N(14)-N(16) 1.41 Å) of the unsubstituted thio-semicarbazides is suggestive of some double-bond character [52,53]. This can be attributed to the presence of terminal aromatic rings, e.g., thiophene, pyrrole attached to the azomethine nitrogen [52,53]. As reported earlier, the C-S single bond distance is 1.82 Å [52] while the double bond is 1.68 Å [54]. In this study, the calculated C–S bond distance of L1 was found to be 1.75 Å. Therefore, the shorter C–S bond of L1 implies the partial double-bond character and this might be due to the resonance involving the thiophene ring. Moreover, the N–H bond distance in both the ligands was found as 1.02 Å [50,55]. 

Compared to the C(1)-C(9)-N(14) angle (~120°) of pyridine-2,6-dicarboxylic dihydrazide [35,50,52,56], shortening of the angles were observed with an average distance of ~113.67° (C(1)-C(10)-N(14)) in both ligands. However, the increased bond angle (~120.47°) at C(10)-N(14)-N(16) in both L1 and L2, compared to the bond angle of pyridine-2,6-dicarboxylic dihydrazide elucidated the change in hybridization to the azomethine nitrogen N(16) for the attachment of an aromatic substituent.

### 4.2. Mulliken Population Analysis

The atomic charges were obtained by the Mulliken population analysis (Appendix A) of both L1 and L2 using the B3LYP level of theory with the 6-311G+(d,p) basis set and the values are tabulated in Table 1. Mulliken population analysis has an important role to predict the polarizability, dipole moment, geometric structure, and bonding capability of a molecule depending on the electronic charge on the coordinating atoms [57,58,59,60]. The charge distribution showing more negative charge was concentrated on O(12) and O(13) of both L1 and L2 as illustrated in Table 1. Thus, the presence of a large negative charge concentration on the oxygen atom is predictive of the oxygen atom to be a primary donor. Additionally, in L1, the observed charge was -0.006 on the N (16) and N (17) atoms, whereas in L2, the value was −0.011 Mulliken on the same atoms. The difference indicates the influence of substituent end groups on the overall ligand charge distribution. The N (44) and N (45) of the pyrrole ring possessed a similar Mulliken charge of −0.126 in L2 while S (42) and S (43) of the thiophene ring carried −0.094 Mulliken in L1.

### 4.3. Vibrational Frequencies 

The experimental (Appendix A) and computed vibrational frequencies of the ligands L1 and L2 with their relative intensities are illustrated in Table 2. Computed IR spectra were calculated using B3LYP/6-311G+(d,p) and B3LYP/cc-pvdz levels of theory in the gas phase. The B3LYP functional can reproduce the experimental vibrational frequencies; however, scaling factors (0.9613 to 0.9688) are required for different basis sets [61]. Furthermore, the calculated vibrational wavenumbers were scaled down using the scaling factor 0.9688 to offset the systematic error caused by neglecting anharmonicity and electron density. The agreement between the experimental and theoretical frequencies is quite good in this study. Some of the differences between these frequencies result from the use of gas phase molecules in the DFT calculations.

In general, the C=O stretching vibrations in the FTIR spectrum occur strongly in the 1870–1540 cm^−1^ region depending on some effects such as substituent, conjugation, and inter- or intra-molecular hydrogen bonding [31,62,63,64]. The strong absorption band at 1698 cm^−1^ in the FTIR spectrum was assigned to the C=O stretching vibration of L1 due to the conjugation with the –NH–N= group and the calculated IR frequency for this vibration was found to be 1760 cm^−1^ for L1 in B3LYP/6-311G+(d,p). Applying 0.9688 as a scaling factor, the frequency for the carbonyl band shifted to 1705 cm^−1^ almost near to the experimental values. The result is approximately close to the analogous compounds [65]. For L2, experimental FT-IR showed a very strong carbonyl (νCO) band at 1677 cm^−1^, similar to L1, and the corresponding scaled stretching vibration was theoretically calculated at 1658 cm^−1^.

The observed band at 1589 cm^−1^ for L1 in the FTIR spectrum was due to the stretching vibration of the C=N group. This vibration band was calculated at 1587 cm^−1^ according to the B3LYP method with the 6-311G+(d,p) basis set. However, for L2, the experimental and theoretical C=N stretching vibrations of the azomethine group as experimentally and theoretically determined appeared at 1550 and 1568 cm^−1^, respectively. The data agree well with the values available in the literature [9,66]. 

The N–H stretching vibration of the secondary amines was obtained in the region of 3500–3000 cm^−1^ in the IR absorption spectra [67]. According to this study, the N–H stretching bands for scaled-B3LYP and experimental FT-IR of L1 were obtained at 3422 and 3318 cm^−1^, respectively. Similarly, the experimental and scaled theoretical absorption bands at 3350 and 3414 cm^−1^ were found for L2. In the FTIR spectrum of L2, the N–H stretching vibration of pyrrole (*ʋ*N–H) was observed at 3469 cm^−1^, whereas it was calculated as 3508 cm^−1^ (scaled). According to the literature, the peaks from 3300 cm^−1^ to 3600 cm^−1^ were assigned to N–H stretching vibrations of the pyrrole ring [68,69].

The characteristic aromatic C–H stretching vibrations appeared in the range 3150–2900 cm^−1^ [70,71,72]. In this work, the weak IR band at 3067 cm^−1^ was identified experimentally for the stretching vibration of aromatic C–H (for thiophene and benzene) of L1, and the relevant vibration was calculated in the range 3139–3084 cm^−1^ for the B3LYP/6-311G+(d,p) level of theory. The bands observed at 994 and 848 cm^−1^ for L1 in FT-IR spectra were assigned to the C–H out-of-plane bending vibration [9,71]. In L2, the observed band 3089 cm^−1^ was attributed to aromatic C–H stretching vibrations. This stretching frequency was calculated at 2921 cm^−1^ using the equivalent calculation method. Likewise, in L1, a pair of bands was observed at 999 and 834 cm^−1^, in L2, which were due to the C–H out-plane skeletal vibration of the pyridine and pyrrole. These are in a good agreement with computed values of 950 and 848 cm^−1^, respectively, by B3LYP/cc-pvdz.

The C=C stretching vibrations of thiophene and pyridine rings appeared experimentally in the range 1540–1505 cm^−1^ [31,73], while the theoretical values were in the region of 1580–1520 cm^−1^ for L1. For L2, the bands observed at 1568 and 1519 cm^−1^ in the FTIR spectrum have been assigned to C=C stretching vibrations of pyridine and pyrrole rings, whereas the theoretically computed values were at 1557 and 1544 cm^−1^ by B3LYP/6-311G+(d,p), which are almost close to experimental values. The C-S-C stretching vibration of the thiophene ring was observed at 727 cm^−1^ experimentally for L1 [31], and this vibration was calculated at 721 cm^−1^ using B3LYP with the 6-311G+(d,p) basis set.

### 4.4. Nuclear Magnetic Resonance (NMR)

NMR is a unique technique to determine the structure of organic compounds. ^1^H and ^13^C-NMR chemical shifts of both L1 and L2 calculated using the GIAO method and CPCM model in DMSO as the solvent are listed in Appendix A and shown in Figure 2. Finally, the calculated results are correlated with the experimental data presented in the synthesis section.

#### 4.4.1. ^1^H-NMR

The hydrazone (NHN=C) protons appeared as singlets in the experimental spectra of L1 and L2 at δ11.43 and 11.42 ppm, respectively, in DMSO-d_6_, which are consistent with the values reported for analogous polytopic ligands [74,75]. For the same protons, resonance signals were computed at δ10.77 and 10.44 ppm, respectively, in DMSO by the B3LYP with 6-311G+(2d,p). The ^1^H-NMR spectra of L2 showed the presence of a singlet at δ11.34 ppm for NH protons of pyrrole [74] and validated well with the calculated value at δ10.18 ppm.

Usually, the chemical shifts of aromatic protons in organic molecules appear in the range δ7.00–8.00 ppm [76,77] and methyl protons (–CH_3_) attached to -N=CR- resonate around δ2.50 ppm [9]. The resonance signals for pyridine protons of the ligand L1 were obtained as a set of multiplets in the range δ8.27–8.35 ppm in the experimental ^1^H-NMR spectrum, while the chemical shifts of these protons were theoretically observed between δ8.85 and 8.43 ppm (in DMSO). The CH protons of the thiophene ring were experimentally observed as a triplet at δ7.14 ppm and two doublets at δ7.61 and 7.66 ppm, whereas theoretically the signals were calculated at δ7.30 ppm for 38/40-H, δ7.66 ppm for 39/41-H, and 7.77 ppm for 22/27-H. For L1, the ^1^H-NMR chemical shift of the –CH_3_ appeared as a singlet at δ2.52 ppm. The calculated chemical shifts of the –CH_3_ were found at δ2.31 ppm for 32/36-H, 2.31 ppm for 33/37-H, and 2.79 ppm for 31/35-H. 

Likewise, for L2, the chemical shift of the pyridine protons was experimentally observed as a set of multiplets at δ8.27–8.34 ppm whereas those were computed at δ8.29–8.62 ppm (in DMSO). Theoretically, the resonance of the signals for 39/41-H, 38/40-H, and 22/27-H protons of the pyrrole ring were found at δ7.17, 6.49, and 6.89 ppm, respectively, while the chemical shift values for these protons were noted in the range δ6.16–6.93 ppm by experiment. Similarly, for L1, the signal observed as a singlet at δ2.39 ppm corresponds to –CH_3_ protons of L2 (Figure 2), while the computed values for the same protons were found at δ2.23–2.63 ppm. Overall, the recorded ^1^H-NMR chemical shifts of both L1 and L2 demonstrate very good agreement with the calculated chemical shifts.

#### 4.4.2. ^13^C-NMR

Ten signals for nineteen carbon atoms are revealed in the experimental ^13^C-NMR spectra for the compounds L1 and L2. The ^13^C-NMR chemical shifts for analogous aromatic organic compounds are usually greater than δ100 ppm [73]. The calculated chemical shifts were observed at δ163.51 and 164.23 ppm for L1 and L2, respectively, while the computed values were at δ159.44 and 159.11 ppm and agree well with the literature [4]. The signals of the azomethine functionality were detected at δ143.22 and 149.25 ppm in the experimental spectra of L1 and L2 [4], whereas the calculations demonstrated the signals at δ154.35 and 155.04 ppm, respectively. The –CH_3_ carbon (30/34-C) showed signals experimentally at δ15.32 and 14.82 ppm for both the compounds and the calculated shifts were found at δ12.36 and 12.05 ppm in DMSO [4,31]. Furthermore, the carbons in pyridine rings of both L1 and L2 appeared at δ148.91, 125.75, 153.65, 125.26, and 140.31 ppm for 1/5-C, 2/4-C, and 3-C, respectively. The rest of the ^13^C signals obtained from thiophene and pyrrole moieties were also detected at the expected regions. In general, the experimental results of both ^1^H and ^13^C-NMR spectra represented a good approximation to the data found theoretically by DFT/6-311G+(d,p).

### 4.5. Thermodynamic Properties of the Reactions

The thermodynamic properties of both the reactions with their dipole moments were calculated in the gas phase and methanol by employing B3LYP/6-311G+(d,p), and the results are presented in Table 3. In addition, the optimized geometries of all the reactants and products involved in chemical reactions are shown graphically in Figure 3. The positive values of the changes of Gibbs free energy (*∆G*), enthalpy (*∆H*), and entropy (*∆S*) of reactions show that the reactions were thermodynamically endothermic. Compared to the changes of *∆G*, *∆H,* and *∆S* profiles in the gas phase, there was a reduction in values in methanol due to the solvent effect on the stabilization of the products. It can be seen from Table 3 that the dipole moments of both L1 and L2 increase from the gas phase to methanol, because the effect of the solvent raises the dipole moments in the molecules due to the increase in the delocalized charge [31]. In general, a higher value of the dipole moment indicates the higher polar nature of a molecule. The value of the molecular dipole moment of L1 seemed to be relatively higher, indicating relatively higher polarity than L2. This parameter is a good indicator to understand the drug–receptor interaction and plays a significant role for the formation of the hydrogen bond in biological systems [78].

### 4.6. Frontier Molecular Orbital (FMO)

The highest occupied molecular orbital (HOMO) and lowest unoccupied molecular orbital (LUMO) are the terms used to describe FMOs that play a crucial part in predicting the chemical stability of the molecule. Moreover, the energies of FMOs are important to determine chemical reactivity where the HOMO represents the ability to donate an electron and LUMO to accept. The energy gap between HOMO and LUMO predicts the kinetics, chemical stability, optical polarizability, and chemical hardness–softness of the molecule [79,80]. The energy gap between HOMO and LUMO can be a pivotal determinant to find a relationship between a class of drug’s activity and their electronic configuration, as reported by Snyder et al. [80]. A larger FO gap corresponds with high kinetic stability, but low chemical reactivity, as it is energetically unfavorable for an electron to elevate from a low-energy HOMO to a relatively high-lying LUMO [81].

Molecular orbital calculation was performed with the optimized structure of the compounds using the B3LYP method with 6-311G+(d,p). The pictographical presentation of all FMOs is displayed in Figure 4 for the title molecules in a gaseous phase. The results revealed that L1 contains 107 occupied and 624 unoccupied molecular orbitals, whereas L2 contains 99 occupied and 629 unoccupied virtual molecular orbitals. For L1, the HOMO of the π-type of the pyrrole moiety lay at -0.226 Hartee (−6.17eV), while the LUMO was the π*-type lying at -0.087 Hartee (−2.38 eV). As a result, a very small energy gap (3.56 eV) was observed between the HOMO and LUMO of L1. Hence, the probability of π → π* electron transition was highly possible in between HOMO and LUMO for the molecule. A similar result was found for the L2. Thus, the FMO energy gap in L2 was found to be 0.131 Hartee (3.56 eV), which was lower than L1, hence enhanced softness, least hardness, and high chemical reactivity (Table 4) are expected. Figure 5 shows that the HOMO of both ligands L1 and L2 was localized on the thiophene/pyrrole ring and the hydrazone group, while LUMO was concentrated on the pyridine ring and the hydrazone group.

To predict the chemical reactivity descriptor of the ligands, molecular orbital calculations were performed at the same level of theory. Considering Parr and Pearson interpretation [82,83,84] of DFT and Koopmans theorem [85], hardness (*η*) and softness (*S*) of both compounds were calculated and are tabulated in Table 4 from the energies (*ε*) of frontier HOMOs and LUMOs according to the following equation [86]:*η* = [*ε*LUMO − *ε*HOMO]/2(1)
*S* = 1/*η*(2)

Using the HOMO and LUMO orbital energies, the ionization energy (I), electron affinity (A), chemical potential (μ), and electrophilicity index (*ω*) of a compound can be calculated as:I = −*ε*HOMO; A = −*ε*LUMO; μ = (*ε*HOMO + *ε*LUMO)/2; *ω* = μ^2^/2*η*(3)

The energies of 6.17 and 5.79 eV were required to ionize an electron from the HOMO of L1 and L2, respectively (Table 4), whereas the energies of 2.38 and 2.23 eV were required to form bonds for L1 and L2, respectively, which indicated their electron acceptance property. The *ω* values of both ligands were calculated as 4.82 and 4.52 eV, respectively. A good electrophile was characterized by a high value of μ, *ω* [87]. Thus, L1 is a better electrophile or less nucleophilic in comparison to L2.

### 4.7. Molecular Electrostatic Potential (MEP)

MEP is related to the total charge distribution of the molecule and is a very useful descriptor in understanding the reactive sites for the electrophilic and nucleophilic attack in chemical reactions as well as hydrogen bonding interactions [88,89]. Thus, it provides a visual understanding of the relative polarity of the molecule. Electrostatic potential surfaces have been plotted for both L1 and L2 by B3LYP/6-311G+(d,p) and these are illustrated in Figure 5. The different values of the electrostatic potential increase in the order red < orange < yellow < green < blue [90,91]. From Figure 5, it was observed that the negative electrostatic potential was located at a maximum, over the oxygen atoms, of the hydrazone group (-CO-NH-), indicating a possible site for electrophilic attack. However, the positive electrostatic potential was crowded over the hydrogen atoms of the same group, which would predict a preferential attack by a nucleophile at that region of the title molecule.

### 4.8. Antimicrobial Activity Using the Agar Disc Diffusion Method

In vitro sensitivities of two Gram-positive and two Gram-negative bacteria including two fungal strains against the synthesized compounds were evaluated by the agar disc diffusion method. The formation of the diameter of inhibition zones in mm by the synthesized analogues are shown in Table 5. Compound L2 showed moderate activity against *Bacillus megaterium* bacteria while L1 showed promising antifungal activity against *Aspergillus niger* fungal strains compared to standard amphotericin-B.

### 4.9. Molecular Docking Study

Molecular docking is a powerful tool to investigate and provide a proper understanding for ligand receptor interactions in order to facilitate the design of potential drugs [92,93,94,95]. To investigate and compare the antimicrobial activity of the synthesized compounds, docking analyses of L1 and L2 against tyrosinase from *Bacillus megaterium* were performed. It is well-known that the tyrosinase of *Bacillus megaterium* bacteria is an attractive target for the development of antimicrobials or antibiotic adjuvants for the treatment of hyperpigmentation because of its similarity (33.5%) to the human enzyme [96,97,98,99]. The docking results were compared with well-testified inhibitor arbutin [97].

#### Binding Affinity of L1 and L2

The highest antibacterial activity (zone of inhibition 12 mm) of compound L2 was detected with tyrosinase from *Bacillus megaterium* (PDB ID: 4j6u) bacteria compared to L1. The binding energies for L1 and L2 with *Bacillus megaterium* were −7.7 and −8.8 kcal mol^−1^, respectively, whereas for arbutin-4j6u the value was −9.1 kcal mol^−1^; the values were calculated by AutoDock Vina. The interactions of the 4j6u with compounds L1 and L2 are shown in the Figure 6.

It was observed that arbutin formed six conventional hydrogen bonds-with 4j6u (Appendix A) by the following residues: ALA40A (one O-H----O-C hydrogen bond), Glu141A (four O-H----O-C hydrogen bond), and LYS47B (one O-H----O-C hydro-gen bond). In addition, several hydrophobic interactions were found with ILE139A, ILE39A, and ALA40A. In L1-4j6u, one conventional hydrogen bond (3.04 Å) of O-H----O-C was observed between O-H of Tyr267A and O-C group of compounds L1. Pi-cation, pi-sulfur, and amide-pi bonds were also noted with LYS47A, PHE48A, ILE39A, ALA40A, GLY43B, and ALA44B. Moreover, ALA44A, LYS47A, ALA44B, LYS47B, PRO52A, ALA40B, and ILE139B were actively involved in the non-covalent interaction (hydrophobic pi-alkyl). L2-4j6u complex was stabilized by four NH….O hydrogen bonds and they were LYS47A (2.25 Å), GLY143B (3.04 Å), Tyr267A (3.07 Å), and PRO219B (2.91 Å) (Figure 6). Like L1, L2 formed pi-cations and amide-pi bonds with LYS47A, ILE39A, and GLY43B, where the distances were 3.73, 4.34, and 3.54 Å. L2 also formed seven pi-alkyl bonds with ALA44A (5.19 Å), LYS47A (4.35 Å), ALA44B (3.95 Å), LYS47B (4.98 Å), PRO52A (5.04 Å), ALA40B (4.67 Å), and ILE139B (4.24 Å), respectively. Results of docking studies revealed that L1 and L2 formed bonds to the active site of tyrosinase and showed strong interactions with Tyr267A, Ala40A, Ala44A, ALA44B, and Lys47B of the tyrosinase enzyme (PDB ID: 4j6u), which are in close vicinity to the control arbutin and support the literature [100,101,102].

Thus, computational results are in good agreement with the in vitro antibacterial behavior of our compounds for novel antibacterial drug design.

## 5. Conclusions

Pyrrole and thiophene as organic molecules and their metal cluster derivatives have been recognized to present a wide range of biological activities in recent years. In the present study we synthesized two tritopic dihydrazide-based ligands bearing pyrrole and thiophene as end groupings and characterized them successfully by FT-IR, ^1^H, and ^13^C-NMR and mass spectrometry. Based on the DFT calculations, a complete structural detail, vibrational, electrostatic potential, Mulliken population, HOMO-LUMO, and thermodynamic analysis was also performed. The computed FT-IR analysis as well as the ^1^H and ^13^C-NMR using the B3LYP/CC-PVDZ/6-311G+(d,p) method agreed satisfactorily with the experimental results. We further evaluated the thermodynamic parameters *∆H*, *∆S*, and *∆G* of the ligands. The geometry optimization revealed the planarity of the L1 and L2 molecules. Further, it was seen from the HOMO-LUMO energy values that the chemical potentials were negative and the frontier orbital gap of the molecule under investigation was small, and hence both compounds are reactive and polarizable. To further showcase the biological activity of the ligands against organic pathogens, the antimicrobial assay was performed and revealed significant inhibition of L2 against *Bacillus*
*megaterium* Gram-positive bacteria, and L1 against *Escherichia coli* and *Aspergillus niger*, although to a lesser extent. The moderate activity of the L2 molecule against *Bacillus*
*megaterium* was substantiated by a molecular docking study against tyrosinase from *Bacillus megaterium* and was found to be significant with a binding energy of -8.8 kcal/mol and three hydrogen bond interactions, which might suggest the antimicrobial activity of the molecule. Overall, L1 and L2 compounds have spurred significant interest for us from the synthetic, computational, and biological points of view. We anticipate continued research regarding these classes of exciting organic ligands.

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
