# Peer review of "Synthesis of Novel Tritopic Hydrazone Ligands: Spectroscopy, Biological Activity, DFT, and Molecular Docking Studies"

_molecules, 2022, doi:10.3390/molecules27051656_

Round 1

Reviewer 1 Report

The manuscript by Sharmin Akther Rupa et al, the manuscript has many serious concerns on the basis that it can not be accepted in this journal.  for example 13 CNMR must be 100 MHz not 400 MHz , value must be either ascending order or descending order (13C NMR (400 MHz,
DMSO-d6, ppm): 14.82 (-CH3), 149.25 (-C=N-), 153.65 (Ar-C), 125.26 (Ar-C), 140.31 (Ar-C),159.11 (C=O), 130.04 (Ar-C), 112.83 (Ar-C), 109.34 (Ar-C), 123.05 (Ar-C). above example showed the manuscript not written carefully

Second one that is very important for this paper: The activity of L2 molecule against Bacillus Megaterium (PDB 4j6u) was significant with a binding energy of -8.8 kcal/mol and three hydrogen bond interactions, which reveals the antimicrobial activity of the molecule. All  both compounds are biologically active but their activity was moderate which did not support their efforts.

third one their binding study:1. Why did the authors consider Tyrosinase from Bacillus megaterium N205A mutant pdb id 4j6u? As they did not report any enzyme specific inhibition related experiments.  2. Docking is a preliminary experiment. How did the author validate the docking protocol?The authors are suggested to use some decoy ligands and calculate the enrichment value in order to justify the docking protocol followed by ALA scan (Alanine scanning)  based MD experiments..  In addition authors are suggested to report at list 100ns Molecular dynamics (MD) simulation guided protein-ligand stability report followed by either MM-GB or PB-SA based thermo data analysis. 3. The pdb id 4j6u does not contain any bound ligand. How did the author select the binding site? If a few sets of amino acids were considered to define the binding site then on what basis they selected them? 4. The authors did not report any Tyrosinase enzyme inhibition assay, they are requested to justify Tyrosinase inhibition as the probable mechanism of action for  these sets of ligands by providing suitable literature references.  

Author Response

Manuscript ID #: Molecules 1583723

Journal: Molecules

A detailed point-by-point response to the reviewers' comments

We would like to thank the reviewers for the comments and endorsements of our work. Attached below is our detailed response to the comments on our manuscript. We have attached a revised version of the paper that addresses the issues raised by the reviewers. The revised parts in the manuscript are indicated by red characters.

Reviewer #1:

  1. Comment:

            The manuscript by Sharmin Akther Rupa et al, the manuscript has many serious concerns on the basis that it cannot be accepted in this journal.  for example, 13C NMR must be 100 MHz not 400 MHz, value must be either ascending order or descending order (13C NMR (400 MHz, DMSO-d6, ppm): 14.82 (-CH3), 149.25 (-C=N-), 153.65 (Ar-C), 125.26 (Ar-C), 140.31 (Ar-C),159.11 (C=O), 130.04 (Ar-C), 112.83 (Ar-C), 109.34 (Ar-C), 123.05 (Ar-C). above example showed the manuscript not written carefully.

Response:

We appreciate the recommendation and comments. We corrected the above-mentioned correction as suggested by the reviewer and presented as follows,

1H NMR (400 MHz, DMSO-d6, ppm): δ 11.43 (s, 2H), 8.34 (m, 2H), 8.28 (m, 1H), 7.65 (d, 2H, J = 4.8 Hz), 7.61 (d, 2H, J = 3.2 Hz), 7.14 (t, 2H, J = 8.8 Hz), 2.51 (s, 6H). 13C NMR (100 MHz, DMSO-d6, ppm): 159.44 (C=O), 154.55 (Ar-C), 148.91 (Ar-C), 143.22 (-C=N-), 140.48 (Ar-C), 130.06 (Ar-C), 129.44 (Ar-C), 128.20 (Ar-C), 125.75 (Ar- C), 15.32 (-CH3).

  1. Comment:

Second one that is very important for this paper: The activity of L2 molecule against Bacillus Megaterium (PDB 4j6u) was significant with a binding energy of -8.8 kcal/mol and three hydrogen bond interactions, which reveals the antimicrobial activity of the molecule. Both compounds are biologically active, but their activity was moderate which did not support their efforts.

Response:

We appreciate the recommendation and agree with the reviewer that the computational and experimental results of antimicrobial activity differ to some extent in this part of the study and required further investigation to understand why such activity differs to the ligands by other means in the future. In this study, we successfully synthesized the novel ligands and studied its spectroscopy with other probable studies. Fortunately, we have found outstanding results during the chemo-sensor study and hopefully, we could be able to show the excellent usage of the ligands as a chemo-sensor in near future. Also, we are planning to do a fluorescence imaging study of protein-ligands.

  1. Comment:

Third one their binding study: 1. Why did the authors consider Tyrosinase from Bacillus megaterium N205A mutant pdb id 4j6u? As they did not report any enzyme specific inhibition related experiments.  2. Docking is a preliminary experiment. How did the author validate the docking protocol? The authors are suggested using some decoy ligands and calculating the enrichment value in order to justify the docking protocol followed by ALA scan (Alanine scanning) based MD experiments. In addition, authors are suggested to report at list 100ns Molecular dynamics (MD) simulation guided protein-ligand stability report followed by either MM-GB or PB-SA based thermo data analysis. 3. The pdb id 4j6u does not contain any bound ligand. How did the author select the binding site? If a few sets of amino acids were considered to define the binding site, then on what basis they select them? 4. The authors did not report any Tyrosinase enzyme inhibition assay, they are requested to justify Tyrosinase inhibition as the probable mechanism of action for these sets of ligands by providing suitable literature references.  

Response:

We appreciate the comments of the reviewer and the following are the answers in response to the question for the kind consideration of the reviewer-

Tyrosinase is a key enzyme in melanogenesis, which is essential for pigmentation. Dysfunction of tyrosinase may cause skin cancer. It is well-known that tyrosinase of Bacillus Megaterium bacteria is an attractive target for the development of antimicrobials or antibiotic adjuvants for the treatment of hyperpigmentation because of its similarity (33.5%) to the human enzyme [97-103]. That’s why, to investigate and compare the antimicrobial activity of the synthesized compounds with in vitro data, docking analysis of L1 and L2 against tyrosinase from Bacillus Megaterium was performed. Also, molecular docking with other proteins of B. Megaterium was conducted, but 46ju provided better binding affinity with the synthesized compounds. For this reason, we aim to study the structure of tyrosinase from the bacteria.

Preliminary antimicrobial studies were carried out to validate the docking protocol. At present, we are doing the fluorescence activities of these ligands and L2 showed excellent fluorescent emission at λmax 520 nm. So, we hope that we will carry out ALA scan (Alanine scanning) based MD experiments and 100ns Molecular dynamics (MD) simulation guided protein-ligand stability in the future for our next study since within this shortest time, we are unable to do these experiments.

The active binding site of tyrosinase was predicted by CASTp. The binding site residues predicted by CASTp for tyrosinase were used for grid generation. We have included this in the manuscript as follows:

4.9. Molecular Docking Study

Molecular docking is a powerful tool to investigate and provide a proper understanding for ligand receptor interactions in order to facilitate the design of potential drugs [93-96]. To investigate and compare the antimicrobial activity of the synthesized compounds, docking analysis of L1 and L2 against tyrosinase from Bacillus Megaterium were performed. It is well-known that tyrosinase of Bacillus Megaterium bacteria is an attractive target for the development of antimicrobials or antibiotic adjuvants for the treatment of hyperpigmentation because of its similarity (33.5%) to the human enzyme [97-100].

4.9.1. Binding affinity of L1 and L2

The highest anti-bacterial activity (zone of inhibition 12 mm) of compound L2 was detected with tyrosinase from Bacillus Megaterium (PDB ID: 4j6u) bacteria compared to L1. The binding energies for L1 and L2 with Bacillus Megaterium were −7.7 and −8.8 kcal mol−1, respectively, which were calculated by AutoDock Vina. The interactions of the 4j6u with compounds L1 and L2 are shown in the Fig. 6.

In L1-4j6u, one conventional hydrogen bond (3.04 Å) of O-H----O-C observed between O-H of Tyr267A and O-C group of compounds L1.  Pi-cation, pi-sulfur and amide-pi bonds were also noticed with LYS47A, PHE48A, ILE39A, ALA40A, GLY43B and ALA44B, respectively. Moreover, ALA44A, LYS47A, ALA44B, LYS47B, PRO52A, ALA40B and ILE139B were actively involved in the non-covalent interaction (hydrophobic pi-alkyl). L2-4j6u complex was stabilized by four NH….O hydrogen bonds and they were LYS47A (2.25 Å), GLY143B (3.04 Å), Tyr267A (3.07 Å) and PRO219B (2.91 Å) (Fig. 6). Like L1, L2 formed pi-cation and amide-pi bonds with LYS47A, ILE39A, GLY43B, where the distances were 3.73, 4.34, 3.54 Å. L2 also formed seven pi-alkyl bonds with ALA44A (5.19 Å), LYS47A (4.35 Å), ALA44B (3.95 Å), LYS47B (4.98 Å), PRO52A (5.04 Å), ALA40B (4.67 Å) and ILE139B (4.24 Å), respectively. Results of docking studies revealed that L1 and L2 formed bonds to the active site of tyrosinase and showed strong interactions with Tyr267A, Ala44A, and ALA44B of tyrosinase enzyme (PDB ID: 4j6u), which also supports the literature [101-103].

Thus, computational results are in good agreement with in vitro antibacterial behaviour of our compounds for novel antibacterial drug design.

We also revise the whole manuscript to make the language and grammar better as suggested by the reviewer.

Reviewer 2 Report

Abedin et al. explored "Synthesis of Novel Tritopic Hydrazone Ligands: Spectroscopy, Biological activity, DFT, and Molecular docking Studies". The manuscript is well-written and must be accepted in its present form for publication in Molecules.

Author Response

Comment:

Abedin et al. explored "Synthesis of Novel Tritopic Hydrazone Ligands: Spectroscopy, Biological activity, DFT, and Molecular docking Studies". The manuscript is well-written and must be accepted in its present form for publication in Molecules.

Response:

We are delighted and thankful for the recommendation.

Reviewer 3 Report

The paper need more attention from the side of the authors.

- The abstract is somehow unorganized. Even if it is an unstructured abstract, it should respond to the following point one after the other. Background, aims, methods, results, and conclusion

- The background is missing.

- The aims are not clearly presented.

- Methods should be written in brief before presenting the results.

- Results they took the most of your abstract. Please be brief; most of the text is not important for a reader who only wants to get an idea about your paper before reading it. Just put your key results in brief.

- The conclusion is missing.

- Font of the reference is different from the text

- This sentence “‪Polytopic ligands containing hydrazide-hydrazone moiety (—CO—NHN=CH—) are important for new drug development” should be better connected to the next one to explain why they are important for new drug development

- Replace “‪In addition to that, the compounds were tested in vitro bioassays against some Gram-negative and Gram-positive ‪bacteria and the fungus strain showing specially promising results for L2. Molecular docking methodology was used to study molecular behavior of L1 and L2 with Bacillus Megaterium to identify their binding interactions” with “‪In addition to that, the compounds were tested in vitro against Gram-negative and positive ‪bacteria and two fungi. Molecular ‪docking was used to study the molecular behavior of L1 and L2 against tyrosinase from Bacillus megaterium.‪

Section 2.3.1.

Reference for the used strains is required. Why are those specific strains used?

Section 3.2.  and 4.9.

ODB ID 4j6u is for tyrosinase from Bacillus megaterium, not the bacillus megatherium cristal structure. Please change the info accordingly

Section 3.2.

The data at least should be split into 2 more subheads, one for the ligand and receptor preparation and the second for the docking analysis. Divide the data into those 2 subheads and add more about the software used to perform the docking and the parameters used (Grid box, extensiveness …)

Section 4.8.

There is no proper description of the results and no discussion in this part.

Please revise

- This sentence should be moved to the materials and methods or deleted from the section “‪Two gram-positive Staphylococcus aureus (cars-2) and Bacillus Megaterium (BTCC-18),

‪two gram-negative Escherichia coli (carsgn-2) and Salmonella Typhi (JCM-1652) bacteria,

‪and two fungal strains Trichoderma harzianum (carsm-2) and Aspergillus niger (carsm-3)

‪were used in this study.

- The molecular docking study is missing a control to properly analyse the results.

- Conclusion: Repeated abstract.  The conclusion must remind the reader why the article was written in the first place and why it is important in the field. The conclusion should briefly give an insight into the obtained results and also the limitations.

Author Response

Manuscript ID #: Molecules 1583723

Journal: Molecules

A detailed point-by-point response to the reviewers' comments

We would like to thank the reviewers for the comments and endorsements of our work. Attached below is our detailed response to the comments on our manuscript. We have attached a revised version of the paper that addresses the issues raised by the reviewers. The revised parts in the manuscript are indicated by red characters.

Reviewer Comment:

  1. The paper need more attention from the side of the authors.

- The abstract is somehow unorganized. Even if it is an unstructured abstract, it should respond to the following point one after the other. Background, aims, methods, results, and conclusion

- The background is missing.

- The aims are not clearly presented.

- Methods should be written in brief before presenting the results.

- Results they took the most of your abstract. Please be brief; most of the text is not important for a reader who only wants to get an idea about your paper before reading it. Just put your key results in brief.

- The conclusion is missing.

Response:

We appreciate the recommendation and rewrite the abstract as follows,

Polytopic organic ligands with hydrazone moiety are in the forefront of new drug research among many others due to their unique and versatile functionality and ease of strategic ligand design. Quantum chemical calculations of these polyfunctional ligands can be carried out in silico to determine the thermodynamic parameters. In this report two new tritopic dihydrazide ligands, N'2, N'6-bis[(1E)-1-(thiophen-2-yl) ethylidene] pyridine-2, 6-dicarbohydrazide (L1) and N'2, N'6-bis[(1E)-1-(1H-pyrrol-2-yl) ethylidene] pyridine-2, 6-dicarbohydrazide (L2) were successfully prepared by the condensation reaction of pyridine-2, 6-dicarboxylic hydrazide with 2-acetylthiophene and 2-acetylpyrrole. The FT-IR, 1H and 13C NMR as well as mass spectra of both L1 and L2 were recorded and analyzed. Quantum chemical calculations were performed at DFT/B3LYP/cc-pvdz/6-311+ G (d, p) level of theory to study the molecular geometry, vibrational frequencies, and thermodynamic properties including changes of ∆H, ∆S, and ∆G for both the ligands. The optimized vibrational frequency and (1H and 13C) NMR obtained by B3LYP/cc-pvdz/6-311 + G (d, p) showed good agreement with experimental FT-IR and NMR data. Frontier molecular orbital (FMO) calculations were also conducted to find the HOMO, LUMO, and HOMO–LUMO gaps of the two synthesized compounds. To investigate the biological activities of the ligands, L1 and L2 were tested in vitro bioassays against some Gram-negative and Gram-positive bacteria and the fungus strain. In addition, Molecular ‪docking was used to study the molecular behavior of L1 and L2 against tyrosinase from Bacillus megaterium. The outcomes revealed that both L1 and L2 can suppress microbial growth of bacteria and fungi with variable potency. The antibacterial activity results demonstrated the compound L2 to be potentially effective against Bacillus Megaterium with inhibition zones of 12 mm while molecular docking study showed the binding energies for L1 and L2 to be −7.7 and −8.8 kcal mol−1 respectively with tyrosinase from Bacillus megaterium.

- Font of the reference is different from the text

Response:

We appreciate the recommendation and corrected the font of the references.

- This sentence “‪Polytopic ligands containing hydrazide-hydrazone moiety (—CO—NHN=CH—) are important for new drug development” should be better connected to the next one to explain why they are important for new drug development?

Response:

We appreciate the recommendation and corrected the section as below:

Polytopic ligands containing hydrazide-hydrazone moiety (—CO—NHN=CH—) are important for new drug development [1-6]. Because, their polyfunctional nature offer multifarious synthetic ways to derivatize such organic molecules towards suitable and effective drug-receptor interaction. The derivatives of hydrazide-hydrazone moiety specially with heterocyclic system possess a range of biological activities namely, anti-microbial, anti-mycobacterial, antitubercular, anticonvulsant, anticholinesterase [1], antiplatelet, and more importantly antitumor [5,7]. Transition metal complexes derived from such type of ligands have been widely studied since they also demonstrate significant biological and pharmacological properties [8-11].

- Replace “‪In addition to that, the compounds were tested in vitro bioassays against some Gram-negative and Gram-positive ‪bacteria and the fungus strain showing specially promising results for L2. Molecular docking methodology was used to study molecular behavior of L1 and L2 with Bacillus Megaterium to identify their binding interactions” with “‪In addition to that, the compounds were tested in vitro against Gram-negative and positive ‪bacteria and two fungi. Molecular ‪docking was used to study the molecular behavior of L1 and L2 against tyrosinase from Bacillus megaterium.‪

Response:

We appreciate the recommendation and corrected the section as below:

This Section is corrected as “‪In addition to that, the compounds were tested in vitro against Gram-negative and positive ‪bacteria and two fungi. Molecular ‪docking was used to study the molecular behavior of L1 and L2 against tyrosinase from Bacillus megaterium.’’

Section 2.3.1.

Reference for the used strains is required. Why are those specific strains used?

Response:

We appreciate the recommendation and added the required references as below.

We have these specimens of strains available in our lab to study. Therefore, we used. We also updated the standard ciprofloxacin and miconazole with Ceftriaxone and Amphotericin-B and corrected the reference the number of Salmonella Typhi (K-323130) bacteria. We are showing our sincere apology for such mistakes. The changes are added as follows,

2.1.3. Antimicrobial activity assay

In vitro antimicrobial activity of synthesized ligands was evaluated by agar disc diffusion method [40]. Mueller Hinton Agar (MHA) media (HIMEDIA, India) was used as a control medium for testing against bacteria and Potato Dextrose Agar (PDA) media (HIMEDIA, India) was used for fungal strain. After preparation, the MHA and PDA medias were incubated for 24 h and contaminations were checked. After incubation, the test organism was inoculated using sterile cotton bar on media. The sample discs were put gently on pre-inoculated agar plates and aerobically incubated for 24 h at 37 °C for antibacterial and for 48 h at 26 °C for antifungal assay. Dimethyl sulfoxide (DMSO) was used as control. Each disc was loaded with 25 µL of sample solution in DMSO containing 300 µg of synthesized compounds. 10 µL of ceftriaxone and amphotericin-B solutions containing 50 µg each in DMSO were loaded per disc for antibacterial and antifungal assays as positive control, respectively. The diameter of the inhibition zones in mm circling the disc were measured. Two gram-positive Staphylococcus aureus (cars-2) and Bacillus Megaterium (BTCC-18), two gram-negative Escherichia coli (carsgn-2) and Salmonella Typhi (K-323130) bacteria, and two fungal strains Trichoderma harzianum (carsm-2) and Aspergillus niger (carsm-3) were used in this study. 

Section 3.2.  and 4.9.

PDB ID 4j6u is for tyrosinase from Bacillus megaterium, not the bacillus megatherium crystal structure. Please change the info accordingly.

Response:

We appreciate the recommendation, and corrected this information as shown below:

3.2. Protein-ligand Docking

3.2.1. Ligand and Protein preparation:

The structures of L1 and L2 have been fully optimized by using Gaussian 09 software at B3LYP/6-311G+ (d, p) level. The 3D crystal structure of tyrosinase from Bacillus Megaterium (PDB ID: 4j6u; resolution: 2.5Å, Chain A, B) was obtained in pdb format from online RCSB protein data bank (PDB) database. The structure was verified, and an energy minimization was performed with the Swiss-Pdb Viewer software packages (version 4.1.0) [44], since the crystal structure contains a variety of issues related to improper bond order, side chains geometry, and missing hydrogen atoms. Prior to docking, all the heteroatoms and water molecules were removed from the crystal structure using PyMol (version 1.3) software packages [45]. The active binding pocket of tyrosinase was predicted by CASTp—having the highest pocket area and volume are 95.432 Å2 and 137.877 Å3, respectively [46]. The binding site residues predicted by CASTp for tyrosinase were used for grid generation. Both the structures of the proteins and ligands were saved in .pdbqt format by AutoDock Vina (version 1.1.2, May 11, 2011) for docking analysis [47].

3.2.2. Molecular docking Analysis:

The docking calculations were performed using default parameters and 8 docked conformations were generated for both compounds. The energy calculations were done by genetic algorithms. Nonpolar hydrogen atoms, Gasteiger partial charges, rotatable bonds, and grid box with dimensions 66.57 × 58.25 × 84.98 Å3 created on the tyrosinase with the aid of Auto Dock Tools 1.1.2 and spacing of 0.3750 Å. The docked conformation of the respective protein conformer with lowest binding free energy and root mean-square deviation value (RMSD) 0.0 Å was analyzed using PyMOL Molecular Graphics System (version 1.7.4) and Accelrys Discovery Studio 4.1 [49].

4.9. Molecular Docking Study

Molecular docking is a powerful tool to investigate and provide a proper understanding for ligand receptor interactions in order to facilitate the design of potential drugs [93-96]. To investigate and compare the antimicrobial activity of the synthesized compounds, docking analysis of L1 and L2 against tyrosinase from Bacillus Megaterium were performed. It is well-known that tyrosinase of Bacillus Megaterium bacteria is an attractive target for the development of antimicrobials or antibiotic adjuvants for the treatment of hyperpigmentation because of its similarity (33.5%) to the human enzyme [97-100].

Section 3.2. The data at least should be split into 2 more subheads, one for the ligand and receptor preparation and the second for the docking analysis. Divide the data into those 2 subheads and add more about the software used to perform the docking and the parameters used (Grid box, extensiveness …)

Response:

We appreciate the recommendation, and splitted the data into 2 more subheads as, 3.2.1. Ligand and Protein preparation and 3.2.2. Molecular docking analysis. Also, more information was added about the software used as shown below:

3.2. Protein-ligand Docking

3.2.1. Ligand and Protein preparation:

The structures of L1 and L2 have been fully optimized by using Gaussian 09 software at B3LYP/6-311G+ (d, p) level. The 3D crystal structure of tyrosinase from Bacillus Megaterium (PDB ID: 4j6u; resolution: 2.5Å, Chain A, B) was obtained in pdb format from online RCSB protein data bank (PDB) database. The structure was verified, and an energy minimization was performed with the Swiss-Pdb Viewer software packages (version 4.1.0) [44], since the crystal structure contains a variety of issues related to improper bond order, side chains geometry, and missing hydrogen atoms. Prior to docking, all the heteroatoms and water molecules were removed from the crystal structure using PyMol (version 1.3) software packages [45]. The active binding pocket of tyrosinase was predicted by CASTp—having the highest pocket area and volume are 95.432 Å2 and 137.877 Å3, respectively [46]. The binding site residues predicted by CastP for tyrosinase were used for grid generation. Both the structures of the proteins and ligands were saved in .pdbqt format by AutoDock Vina (version 1.1.2, May 11, 2011) for docking analysis [47].

3.2.2. Molecular docking Analysis:

The docking calculations were performed using default parameters and 8 docked conformations were generated for both compounds. The energy calculations were done by genetic algorithms. Nonpolar hydrogen atoms, Gasteiger partial charges, rotatable bonds, and grid box with dimensions 66.57 × 58.25 × 84.98 Å3 created on the tyrosinase with the aid of Auto Dock Tools 1.1.2 and spacing of 0.3750 Å. The docked conformation of the respective protein conformer with lowest binding free energy and root mean-square deviation value (RMSD) 0.0 Å was analyzed using PyMOL Molecular Graphics System (version 1.7.4) and Accelrys Discovery Studio 4.1 [49].

Section 4.8. There is no proper description of the results and no discussion in this part.

Please revise This sentence should be moved to the materials and methods or deleted from the section “‪Two gram-positive Staphylococcus aureus (cars-2) and Bacillus Megaterium (BTCC-18), two gram-negative Escherichia coli (carsgn-2) and Salmonella Typhi (JCM-1652) bacteria, and two fungal strains Trichoderma harzianum (carsm-2) and Aspergillus niger (carsm-3) ‪were used in this study.

Response:

We appreciate the recommendation, and this section is updated by changing the standard ciprofloxacin and miconazole with Ceftriaxone and Amphotericin-B.             Also, the sentences of second part (4.8) are deleted and added to 2.3.1 as suggested by the reviewer.

2.1.3. Antimicrobial activity assay

In vitro antimicrobial activity of synthesized ligands was evaluated by agar disc diffusion method [40]. Mueller Hinton Agar (MHA) media (HIMEDIA, India) was used as a control medium for testing against bacteria and Potato Dextrose Agar (PDA) media (HIMEDIA, India) was used for fungal strain. After preparation, the MHA and PDA medias were incubated for 24 h and contaminations were checked. After incubation, the test organism was inoculated using sterile cotton bar on media. The sample discs were put gently on pre-inoculated agar plates and aerobically incubated for 24 h at 37 °C for antibacterial and for 48 h at 26 °C for antifungal assay. Dimethyl sulfoxide (DMSO) was used as control. Each disc was loaded with 25 µL of sample solution in DMSO containing 300 µg of synthesized compounds. 10 µL of ceftriaxone and amphotericin-B solutions containing 50 µg each in DMSO were loaded per disc for antibacterial and antifungal assays as positive control, respectively. The diameter of the inhibition zones in mm circling the disc were measured. Two gram-positive Staphylococcus aureus (cars-2) and Bacillus Megaterium (BTCC-18), two gram-negative Escherichia coli (carsgn-2) and Salmonella Typhi (K-323130) bacteria, and two fungal strains Trichoderma harzianum (carsm-2) and Aspergillus niger (carsm-3) were used in this study. 

4.8. Antimicrobial activity using agar disc diffusion method

In vitro sensitivities of two gram-positive and two gram-negative bacteria including two fungal strains against the synthesized compounds were evaluated by agar disc diffusion method. The formation of diameter of inhibition zones in mm by the synthesized analogues are shown in Table 5. Compound L2 showed moderate activity against Bacillus Megaterium bacteria while L1 showed promising antifungal activity against Aspergillus niger fungal strains compared to standard Amphotericin-B.

Table 5. Diameter of inhibition zones (mm) of the synthesized compounds, Ceftriaxone and Amphotericin-B against tested bacterial and fungal strains.

Compd.

Gram (+) bacteria

Gram (-) bacteria

Fungi

S. aureus

B. megaterium

E. coli

S. typhi

T. harzianum

A. niger

L1

10

10

11

9

6

11

L2

9

12

10

8

6

6

Ceftriaxone

40.0

50.0

38.0

44.0

Amphotericin-B

17.0

8.0

- The molecular docking study is missing a control to properly analyses the results.

Response:

We appreciate the recommendation and updated the molecular docking study by adding more details and literature references as follows,

4.9. Molecular Docking Study

Molecular docking is a powerful tool to investigate and provide a proper understanding for ligand receptor interactions in order to facilitate the design of potential drugs [93-96]. To investigate and compare the antimicrobial activity of the synthesized compounds, docking analysis of L1 and L2 against tyrosinase from Bacillus Megaterium were performed. It is well-known that tyrosinase of Bacillus Megaterium bacteria is an attractive target for the development of antimicrobials or antibiotic adjuvants for the treatment of hyperpigmentation because of its similarity (33.5%) to the human enzyme [97-100].

- Conclusion: Repeated abstract. The conclusion must remind the reader why the article was written in the first place and why it is important in the field. The conclusion should briefly give an insight into the obtained results and also the limitations.

Response:

We appreciate the recommendation and changed the conclusion as follows,

Pyrrole and thiophene as organic molecules and their metal cluster derivatives have been recognized to present a wide range of biological activities in recent years. In this present study we have synthesized two tritopic dihydrazide based ligands bearing Pyrrole and Thiophene as end groupings and characterized successfully by FT-IR, 1H and 13C NMR and mass spectrometry. Based on the DFT the calculations, a complete structural detail, vibrational, electrostatic potential, Mulliken population, HOMO-LUMO and thermodynamic analysis were also done. The computed FT-IR analysis as well as the 1H and 13C NMR using B3LYP/CC-PVDZ/6-311+G(d, p) method agreed satisfactorily with the experimental results. We further evaluated the thermodynamic parameters ∆H, ∆S, and ∆G of the ligands. The geometry optimization revealed the planarity of L1 and L2 molecules. Further, it was seen from the HOMO-LUMO energy values that the chemical potentials were negative and the frontier orbital gap of the molecule under investigation was small, and hence, both compounds are reactive and polarizable. To further showcase the biological activity of the ligands against organic pathogens, the antimicrobial assay was performed and revealed significant inhibition of L2 against Bacillus Megaterium gram positive bacteria, and L1 against Escherichia coli, Aspergillus niger although in lesser extent. The moderate activity of L2 molecule against Bacillus Megaterium is substantiated by molecular docking study against tyrosinase from Bacillus megaterium and was found significant with a binding energy of -8.8 kcal/mol and three hydrogen bond interactions, which might suggest the antimicrobial activity of the molecule. Overall, L1 and L2 compounds have spurred significant interest for us from the synthetic, computational and biological point of view. We anticipate continued research regarding these classes of exciting organic ligands.

We also revise the whole manuscript to make the language and grammar better as suggested by the reviewer.

Dr. Md Abdul Majed Patwary

Comilla University

Round 2

Reviewer 1 Report

 The manuscript by Sharmin Akther Rupa et al is a comprehensive study on Synthesis of Novel Tritopic Hydrazone Ligands: Spectroscopy,
Biological activity, DFT, and Molecular docking Studies. Pyrrole and thiophene as organic molecules and their metal cluster derivatives have been recognized to
present a wide range of biological activities in recent years. In this present study we
have synthesized two tritopic dihydrazide based ligands bearing Pyrrole and Thiophene as end groupings and characterized successfully by FT-IR, 1H and 13C NMR and massspectrometry. Based on the DFT the calculations, a complete structural detail, vibrational, electrostatic potential, Mulliken population, HOMO-LUMO and thermodynamic analysis were also done
This manuscript is suitable publication in Molecules.

Author Response

Reviewer #1:

Comment:

The manuscript by Sharmin Akther Rupa et al is a comprehensive study on the Synthesis of Novel Tritopic Hydrazone Ligands: Spectroscopy, Biological activity, DFT, and Molecular docking Studies. Pyrrole and thiophene as organic molecules and their metal cluster derivatives have been recognized to present a wide range of biological activities in recent years. In this present study, we have synthesized two tritopic dihydrazide based ligands bearing Pyrrole and Thiophene as end groupings and characterized successfully by FT-IR, 1H and 13C NMR and mass spectrometry. Based on the DFT the calculations, a complete structural detail, vibrational, electrostatic potential, Mulliken population, HOMO-LUMO and thermodynamic analysis were also done.

This manuscript is suitable for publication in Molecules.

Response:

We are delighted and thankful for the recommendation of the reviewer.

Reviewer 3 Report

The effort made by the authors during the first round of revision is clear and the paper was improved greatly. nevertheless some points were not addressed properly.

1. The molecular study needs a control molecule to compare the results of  L1 and L2. how would we know that −7.7 and −8.8 are good scores or not. Either perform a quick analysis for a molecule known for its tyrosinase inhibition or compare with other studies control that used the same methods in your paper.
For exemple in this study "https://doi.org/10.1016/j.bmc.2011.10.078" they used Arbutin to compare the results of their studied molecule.

2. The references are in a weird format containing only the author's name, journal name and date. article title and other data are missing. 
Reference style followed by the journal

Author 1, A.B.; Author 2, C.D. Title of the article. Abbreviated Journal Name Year, Volume, page range.

Author Response

Manuscript ID #: Molecules 1583723

Journal: Molecules

A detailed point-by-point response to the reviewers' comments

We would like to thank the reviewers for the comments and endorsements of our work. Attached below is our detailed response to the comments on our manuscript. We have attached a revised version of the paper that addresses the issues raised by the reviewers. The revised parts in the manuscript are indicated by red characters.

Reviewer:

Comment: The effort made by the authors during the first round of revision is clear and the paper was improved greatly. Nevertheless, some points were not addressed properly.

  1. The molecular study needs a control molecule to compare the results of L1 and L2. how would we know that −7.7 and −8.8 are good scores or not? Either perform a quick analysis for a molecule known for its tyrosinase inhibition or compare with other studies control that used the same methods in your paper. For example, in this study "https://doi.org/10.1016/j.bmc.2011.10.078" they used Arbutin to compare the results of their studied molecule.

Response:

We appreciate the recommendation and rewrote the Molecular Docking study as follows as suggested by the reviewer,

4.9. Molecular Docking Study

Molecular docking is a powerful tool to investigate and provide a proper understanding of ligand-receptor interactions in order to facilitate the design of potential drugs [92-95]. To investigate and compare the antimicrobial activity of the synthesized compounds, docking analyses of L1 and L2 against tyrosinase from Bacillus Megaterium were performed. It is well-known that tyrosinase of Bacillus Megaterium bacteria is an attractive target for the development of antimicrobials or antibiotic adjuvants for the treatment of hyperpigmentation because of its similarity (33.5%) to the human enzyme [96-99]. The docking results were also compared with well-testified inhibitor arbutin [97].

4.9.1. Binding affinity of L1 and L2

The highest antibacterial activity (zone of inhibition 12 mm) of compound L2 was detected with tyrosinase from Bacillus Megaterium (PDB ID: 4j6u) bacteria compared to L1. The binding energies for L1 and L2 with Bacillus Megaterium were −7.7 and −8.8 kcal mol−1, respectively, whereas for arbutin-4j6u the value was -9.1 kcal mol-1 which were calculated by AutoDock Vina. The interactions of the 4j6u with compounds L1 and L2 are shown in Fig. 6.

It was observed that arbutin formed six conventional hydrogen bonds with 4j6u (Supplementary Fig. S11) by the following residues: ALA40A (one O-H----O-C hydrogen bond), Glu141A (four O-H----O-C hydrogen bond), and LYS47B (one O-H----O-C hydro-gen bond). Also several hydrophobic interactions were found with ILE139A, ILE39A, and ALA40A. In L1-4j6u, one conventional hydrogen bond (3.04 Å) of O-H----O-C observed between O-H of Tyr267A and O-C group of compounds L1.  Pi-cation, pi-sulfur and amide-pi bonds were also noticed with LYS47A, PHE48A, ILE39A, ALA40A, GLY43B and ALA44B, respectively. Moreover, ALA44A, LYS47A, ALA44B, LYS47B, PRO52A, ALA40B and ILE139B were actively involved in the non-covalent interaction (hydrophobic pi-alkyl). L2-4j6u complex was stabilized by four NH….O hydrogen bonds and they were LYS47A (2.25 Å), GLY143B (3.04 Å), Tyr267A (3.07 Å) and PRO219B (2.91 Å) (Fig. 6). Like L1, L2 formed pi-cation and amide-pi bonds with LYS47A, ILE39A, GLY43B, where the distances were 3.73, 4.34, 3.54 Å. L2 also formed seven pi-alkyl bonds with ALA44A (5.19 Å), LYS47A (4.35 Å), ALA44B (3.95 Å), LYS47B (4.98 Å), PRO52A (5.04 Å), ALA40B (4.67 Å) and ILE139B (4.24 Å), respectively. Results of docking studies revealed that L1 and L2 formed bonds to the active site of tyrosinase and showed strong interactions with Tyr267A, Ala40A, Ala44A, ALA44B and Lys47B of tyrosinase enzyme (PDB ID: 4j6u), which are in close vicinity to the control arbutin and supports the literature [100-102].

  1. The references are in a weird format containing only the author's name, journal name and date. article title and other data are missing. Reference style followed by the journal

    Author 1, A.B.; Author 2, C.D. Title of the article. Abbreviated Journal Name Year, Volume, page range.

Response:

We appreciate the recommendation and corrected the style of the references as suggested by the reviewer.

Reference:

[1] Kaan, K.; Halise, I.; Parham, T.; Ilhami, G.; Claudiu, T. Investigation of inhibitory properties of some hydrazone compounds on hCA I, hCA II and AChE enzymes. Bioorg. Chem., 2019, 86, 316–321. https://doi.org/10.1016/j.bioorg.2019.02.008.

[2] Naseem, S.; M. Khalid M.; M. N.Tahir M. N.;and M. A. Halim, M. A.; A. A.C. Braga, A. A.C.; M. M. Naseer M. M.; and Z. Shafiq. Synthesis, structural, DFT studies, docking and antibacterial activity of a xanthene-based hydrazone ligand.  J. Mol. Struct., 2017, 1143, 235-244.

[3] Xu, J.; Zhou, T.; Xu, Z.Q.; Gu, X.-N.; Wu, W.N.; Chen, H.; Wang, Y.; Jia, L.; Zhu, T.F.; Chen,.R.H. Synthesis, crystal structures and antitumor activities of copper(II) complexes with a 2-acetylpyrazine isonicotinoyl hydrazone ligand. J. Mol. Struct., 2017, 1128, 448-454.

[4] Bingul, M.; Ercan, S.; Boga, M. The design of novel 4,6-dimethoxyindole based hydrazide- hydrazones: Molecular modeling, synthesis and anticholinesterase activity. J. Mol. Struct., 2020, 1213, 128202. https://doi.org/10.1016/j.molstruc.2020.128202

[5] El-Medani, S. M.; Makhlouf, A. A.; Moustafa, H.; Afifi, M. A.; Haukka M.; Ramadan, R. M. Spectroscopic, crystal structural, theoretical and biological studies of phenylacetohydrazide Schiff base derivatives and their copper complexes.  J. Mol. Struct., 2020, 1208, 127860.

[6] Shebl, M. Coordination behavior of new bis(tridentate ONO, ONS and ONN) donor hydrazones towards some transition metal ions: Synthesis, spectral, thermal, antimicrobial and antitumor studies. J.  Mol. Struct., 2017, 1128, 79-93.

[7] Nasr, T.; Bondock, S.; Youns, M. Anticancer activity of new coumarin substituted hydrazide–hydrazone derivatives. Eur. J. Med. Chem., 2014, 76, 539-548.

[8] Zhao, L.; Xu, Z.; Thompson, L. K.; Heath, S. L.; Miller, D. O.; Ohba,. M. Synthesis, Structure, and Magnetism of a Novel Alkoxide Bridged Nonacopper(II) (Cu9O12) [3×3] Square Grid Generated by a Strict Self-Assembly Process. Angew. Chem., Int. Ed., 2000, 112, 3244-3247.

[9] Pavan, F. R.; Maia, P. I. S.; Leite, S. R. A.; Deflon, V. M.; Batista, A. A.; Sato, D. N.; Franzblau, S. G.; Leite, C. Q. F. Ruthenium (II) phosphine/picolinate complexes as antimycobacterial agents. Eur. J. Med. Chem., 2010, 45, 598-601. https://doi.org/10.1016/j.ejmech.2009.10.049

[10] Zhao, L.; Niel, V.; Thompson, L. K.; Xu, Z.; Milway, V. A.; Harvey, R. G.; Miller, D. O.; Wilson, C.; Leech, M.; Howard J. A. K.; Heath, S. L. Self-assembled polynuclear clusters derived from some flexible polydentate dihydrazide ligands. Dalton Trans., 2004, 1446-1455.      

DOI: https://doi.org/10.1039/B317091H

[11] Dawe, L. N.; Shuvaev, K. V.; Thompson, L. K. Magnetic [n × n] (n = 2−5) Grids by Directed Self-Assembly. Inorg. Chem., 2009, 48, 3323-3341.

[12] Ayme, J. F.; Lehn, J. M.; Bailly, C.; Karmazin, L. Simultaneous Generation of a [2 × 2] Grid-Like Complex and a Linear Double Helicate: a Three-Level Self-Sorting Process.  J. Am. Chem. Soc., 2020, 142 (12), 5819-5824.

[13] Barry, D. E.; Caffrey, D. F.; Gunnlaugsson, T. Lanthanide-directed synthesis of luminescent self-assembly supramolecular structures and mechanically bonded systems from acyclic coordinating organic ligands. Chem. Soc. Rev., 2016, 45, 3244-3274.

[14] Zhao, L.; Xu, Z.; Thompson, L. K.; Heath, S. L.; Miller, D. O.; Ohba, M. Synthesis, Structure, and Magnetism of a Novel Alkoxide Bridged Nonacopper(II) (Cu(9)O(12)). Angew. Chem., Int. Ed., 2000, 39, 3114-3117.

[15] Mandal, T. N.; Roy, S.; Konar, S.; Jana, A.; Ray, S.; Das, K.; Saha, R.; Fallah, M. S. E.; Butcher, R. J.; Chatterjeee, S.; Kar, S. K. Self-assembled tetranuclear Cu4(II), Ni4(II)[2x2] square grids and a dicopper(II) complex of heterocycle based polytopic ligands – Magnetic studies†‡. Dalton Trans., 2011, 40, 11866-11875.

[16] Dawe, L. N.; Abedin, T. S. M.; Thompson, L. K. Ligand directed self-assembly of polymetallic [n × n] grids: rational routes to large functional molecular subunits? Dalton Trans., 2008, 1661-1675.

[17] Dawe, L. N.; Abedin, T. S. M.; Kelly, T. L.; Thompson, L. K.; Miller, D. O.; Zhao, L.; Wilson, C.; Leech, M. A.; Howard, J. A. K. Self-assembled polymetallic square grids ([2 × 2] M4, [3 × 3] M9) and trigonal bipyramidal clusters (M5)—structural and magnetic properties.  J. Mater. Chem., 2006, 16, 2645-2659.

[18] Abedin, T. S. M.; Thompson, L. K.; Miller, D. O. An octanuclear [Co(II)2–Co(III)2]2 interlocked grid example of an inorganic [2]catenane†.  Chem. Commun., 2005, 5512-5514.

[19] Abedin, T. S. M.; Thompson, L. K.; Miller, D. O.; Krupicka, E. Structural and magnetic properties of a self-assembled spheroidal triakonta-hexanuclear Cu36 cluster. Chem. Commun., 2003, 708-709.

[20] Tikhomirov, A. S.; Litvinova, V. A.; Andreeva, D. V.; Tsvetkov, V. B.; Dezhenkova, L. G.; Volodina, Y. L.; Kaluzhny, D. N.; Treshalin, I. D.; Schols, D.; Ramonova, A. A.; Moisenovich, M. M.; Shtil, A. A.; Shchekotikhin, A. E. Amides of pyrrole- and thiophene-fused anthraquinone derivatives: A role of the heterocyclic core in antitumor properties. Eur. J. Med. Chem., 2020, 199, 112294.        

[21] Kundu, T.; Pramanik, A. Expeditious and eco-friendly synthesis of new multifunctionalized pyrrole derivatives and evaluation of their antioxidant property. Bioorg. Chem., 2020, 98, 103734.

[22] Cherif, O.; Agrebi, A.; Alves, S.; Baleizão, C.; Farinha, J. P.; Allouche, F. Synthesis and fluorescence properties of aminocyanopyrrole and aminocyanothiophene esthers for biomedical and bioimaging applications. J.  Mol. Struct., 2020, 1209, 127974.

[23] Keri, R. S.; Chand, K.; Budagumpi, S.; Somappa, S. B.; Patil, S. A.; Nagaraja, B. M. An overview of benzo[b]thiophene-based medicinal chemistry. Eur. J. Medi. Chem., 2017, 138, 1002-1033.

[24] Rackham, M. D.; Brannigan, J. A.; Moss, D. K.; Yu, Z.; Wilkinson, A. J.; Holder, A. A.; Tate, E. W.; Leatherbarrow, R. J. Discovery of novel and ligand-efficient inhibitors of Plasmodium falciparum and Plasmodium vivax N-myristoyltransferase. J. Med. Chem., 2013, 56, 371-375.

[25] Jagtap, V. A.; Agasimundin, Y. S. Synthesis and preliminary evaluation of some 2-amino-N'-[substituted]-4,5,6,7-tetrahydro-1-benzothiophene-3-carbohydrazide as antimicrobial agents. J. Pharm. Res., 2015, 9, 10-14.

[26] Berrade, L.; Aisa, B.; Ramirez, M. J.; Galiano, S.; Guccione, S.; Moltzau, L. R.; Levy, F. O.; Nicoletti, F.; Battaglia, G.; Molinaro, G.; Aldana, I.; Monge, A.; Perez-Silanes, S. Novel Benzo[b]thiophene Derivatives as New Potential Antidepressants with Rapid Onset of Action. J. Med. Chem., 2011, 54, 3086-3090.

[27] Mourey, R. J.; Burnette, B. L.; Brustkern, S. J.; Daniels, J. S.; Hirsch, J. L.; Hood, W. F.; Meyers, M. J.; Mnich, S. J.; Pierce, B. S.; Saabye, M. J.; Schindler, J. F.; South, S. A. Webb, E. G.; Zhang, J.; Anderson, D. R. A benzothiophene inhibitor of mitogen-activated protein kinase-activated protein kinase 2 inhibits tumor necrosis factor alpha production and has oral anti-inflammatory efficacy in acute and chronic models of inflammation. J. Pharmacol. Exp. Ther., 2010, 333, 797-807.

[28] Rao, G. K.; Subramaniam, R. Synthesis, Antitubercular and Antibacterial Activities of Some Quinazolinone Analogs Substituted with Benzothiophene. Chem. Sci. J., 2015, 6, 92-96.

[29] Zaher, A. F.; Khalil, N. A.; Ahmed, E. M.; Synthesis and anticonvulsant activity of new 3′-aryl-7-bromo-spiro[[1]benzothiophene-3,2′-[1,3] thiazolidine]-2,4′-dione derivatives. Ori. J. Chem., 2010, 26, 1241-1248.

[30] Gündüzalp, A. B; Özsen, E.; Alyar, H.; Alyar, S.; Özbek, N. Biologically active Schiff bases containing thiophene/furan ring and their copper (II) complexes: Synthesis, spectral, nonlinear optical and density functional studies. J.  Mol. Struct., 2016, 1120, 259-266.

[31] Ermiş, E. Synthesis, spectroscopic characterization and DFT calculations of novel Schiff base

containing thiophene ring. J.  Mol. Struct., 2018, 1156, 91-104.

[32] Balachandran, V.; Santhi, G.; Karpagam, V.; Lakshmi, A. Molecular structure, spectroscopic (FT-IR, FT-Raman), NBO and HOMO–LUMO analyses, computation of thermodynamic functions for various temperatures of 2, 6-dichloro-3-nitrobenzoic acid. Spectrochim. Acta Part A: Mol. and Biomol. Spect., 2013, 110, 130-140.

[33] Aljahdali, M. S.; Abdou El-Sherif, A.; Hilal, R.H.; Abdel-Karim, A.T. Mixed bivalent transition metal complexes of 1,10-phenanthroline and 2-aminomethylthiophenyl-4-bromo salicyl aldehyde Schiff base: Spectroscopic, molecular modeling and biological activities. Chem. Eur. J., 2013, 4, 370-378.

[34] Balachandran, V.; Santhi, G.; Karpagam, V.; Lakshmi, A. Molecular structure, spectroscopic (FT-IR, FT-Raman), NBO and HOMO–LUMO analyses, computation of thermodynamic functions for various temperatures of 2, 6-dichloro-3-nitrobenzoic acid. Spectrochim. Acta Part A: Mol. and Biomol. Spect., 2013, 110, 130-140.

[35] Naveen, R. K.; Tittal, V.D.; Ghule, N.; Kumar, L.; Lal, K.; Kumar, A. Design, synthesis, biological activity, molecular docking and computational studies on novel 1,4-disubstituted-1,2,3-Triazole-Thiosemicarbazone hybrid molecules J.  Mol. Struct., 2020, 1209, 127951. https://doi.org/10.1016/j.molstruc.2020.127951

[36] Palafox, M. A. DFT computations on vibrational spectra: Scaling procedures to improve the wavenumbers. Phy. Sci. Rev., 2018, 3, 20170184-20170214. https://doi.org/10.1515/psr-2017-0184

[37] James, C.; Raj, A. A.; Reghunathan, R.; Jayakumar, V. S.; Joe, I. H.; Structural conformation and vibrational spectroscopic studies of 2,6‐bis(p‐N,N‐dimethyl benzylidene)cyclohexanone using density functional theory. J. Raman Spectrosc., 2006, 37, 1381-1392.

[38] Furic, K.; Duric, J. R. Proton-pair disorder in dimers of aromatic carboxylic acids: vibrational spectra of benzoic acid at low temperatures. Chem. Phys. Lett., 1986, 126, 92-97.

[39] Thompson, L. K.; Dawe, L. N. Magnetic properties of transition metal (Mn(II), Mn(III), Ni(II), Cu(II)) and lanthanide (Gd(III), Dy(III), Tb(III), Eu(III), Ho(III), Yb(III)) clusters and [nxn] grids: Isotropic exchange and SMM behaviour. Coord. Chem. Rev., 2015, 289–290, 13-31.

[40] Balouiri, M.; Sadiki, M.; Ibnsouda, S. K. Methods for in vitro evaluating antimicrobial activity: A review. J. Pharm. Anal., 2016, 6, 71-79. https://doi.org/10.1016/j.jpha.2015.11.005

[41] Gaussian 09, Revision A.1, Frisch, M.J.; Trucks, G.W.; Schlegel, H.B.; Scuseria, G.E.; Robb, M.A.; Cheeseman, J.R.; Scalmani, G.; Barone, V.; Mennucci, B.; Petersson, G. A.; Nakatsuji, H.; Caricato, M.; Li, X.; Hratchian, H. P.; Izmaylov, A. F.; Bloino, J.; Zheng, G.; Sonnenberg, J. L.; Hada, M.; Ehara, M.; Toyota, K.; Fukuda, R.; Hasegawa, J.; Ishida, M.; Nakajima, T.; Honda, Y.; Kitao, O.; Nakai, H.; Vreven, T.; Montgomery Jr., J. A.; Peralta, J. E.; Ogliaro, F.; Bearpark, M.; Heyd, J. J.; Brothers, E.; Kudin, K. N.; Staroverov, V. N.; Kobayashi, R.; Normand, J.; Raghavachari, K.; Rendell, A.P.; Burant, J.C.; Iyengar, S.S.; Tomasi, J.; Cossi, M.; Rega, N.; Millam, J.M.; Klene, M.; Knox, J. E.; Cross, J. B.; Bakken, V.; Adamo, C.; Jaramillo, J.; Gomperts, R.; Stratmann, R. E.; Yazyev, O.; Austin, A. J.; Cammi, R.; Pomelli, C.; Ochterski, J. W.; Martin, R. L.; Morokuma, K.; Zakrzewski, V. G.; Voth, G. A.; Salvador, P.; Dannenberg, J. J.; Dapprich, S.; Daniels, A. D.; Farkas, Ö.; Foresman, J. B.; Ortiz, J. V;. Cioslowski, J.; Fox, D. J. Gaussian Inc., Wallingford CT, 2009.

[42] Becke, A. D. Density‐functional thermochemistry. III. The role of exact exchange. J. Chem. Phys., 1993, 98, 5648-5652.

[43] Lee, C.; Yang, W.; Parr, R. G. Development of the Colle-Salvetti correlation-energy formula into a functional of the electron density. Phys. Rev. B, 1993, 37, 785-789. https://doi.org/10.1103/PhysRevB.37.785

[44] Guex, N.; Peitsch, M. C.; SWISS-MODEL and the Swiss-Pdb Viewer: an environment for comparative protein modeling. Electrophoresis, 1997, 18(15), 2714-2723.

[45] DeLano, W.L. The PyMOL Molecular Graphics System. 2002, Delano Scientific, San Carlos.

[46] Dundas, J.; Ouyang, Z.; Tseng, J.; Binkowski, A.; Turpaz, Y.; Liang, J. CASTp: computed atlas of surface topography of proteins with structural and topographical mapping of functionally annotated residues. Nucleic Acids Res., 2006, 34, W116-W118.

[47] Trott, O.; Olson, A. J. AutoDock Vina: improving the speed and accuracy of docking with a new scoring function, efficient optimization and multithreading. J. Com. Chem., 2010, 31(2), 455-461.

[48] Accelrys Software Inc., Discovery Studio Modeling Environment, Release 4.0, San Diego: Accelrys Software Inc., 2013.

[49] Grove, H.; Kelly, T. L.; Thompson, L. K.; Zhao, L.; Xu, Z.; Abedin, T. S. M.; Miller, D. O.; Goeta, A. E.; Wilson, C.; Howard, J. A. K. Copper(II) Complexes of a Series of Alkoxy Diazine Ligands: Mononuclear, Dinuclear, and Tetranuclear Examples with Structural, Magnetic, and DFT Studies. Inorg. Chem., 2004, 43 (14), 4278-4288.

[50] Singh, R. N.; Kumar, A.; Tiwari, R. K.; Rawat, P.; Verma, D. Synthesis Molecular Structure and Spectral Analysis o f Ethyl 4-Formyl-3, 5-Dimethyl 1H-Pyrrole-2-Carboxylate Thiosemicarbazone: A Combined DFT and AIM Approach. J.  Mol. Struct., 2012, 1016, 97-108.

[51] Pramodh, B.; Lokanath, N. K.; Naveen, S.; Naresh, P.; Ganguly, S.; Panda, J. Molecular structure, Hirshfeld surface analysis, theoretical investigations and nonlinear optical properties of a novel crystalline chalcone derivative: (E)-1-(5-bromothiophen-2-yl)-3-(p-tolyl)prop-2-en-1-one. J.  Mol. Struct., 2018, 1161, 9-17. https://doi.org/10.1016/j.molstruc.2018.01.078

[52] Jouad, El. M.; Riou, A.; Allain, M.; Khan, M. A.; Boue, G. M. Synthesis, structural and spectral studies of 5-methyl 2-furaldehyde thiosemicarbazone and its Co, Ni, Cu and Cd complexes. Polyhedron, 2001, 20, 67-74.

[53] Guerraoui, A.; Djedouani, A.; Jeanneau, E.; Boumaza, A.; Alsalme, A.; Zarrouk, A.; Salih, K. S. M.; Warad, I. Crystal structure and spectral of new hydrazine-pyran-dione derivative: DFT enol↔hydrazone tautomerization via zwitterionic intermediate, hirshfeld analysis and optical activity studies. J.  Mol. Struct., 2020, 1220, 128728. https://doi.org/10.1016/j.molstruc.2020.128728

[54] Anderson, B. J.; Jasinski, J. P.; Freedman, M.B.; Millikan, S. P.; O’Rourke, K. A.; Smolenski, V. A. Synthesis, Crystal Structural Investigations, and DFT Calculations of Novel Thiosemicarbazones. Crystals, 2016, 6, 17-35. https://doi.org/10.3390/cryst6020017

[55] Uzun, S.; Demircioglu, Z.; Taşdogan, M.; Agar, E. Quantum chemical and X-ray diffraction studies of (E)-3-(((3,4-dimethoxybenzyl)imino)methyl)benzene-1,2-diol. J.  Mol. Struct., 2020, 1206, 127749.

[56] Mahmoudi, F.; Farhadi, S.; Dusek, M.; Poupon, M. Synthesis, Spectroscopy and X-ray Crystallography Structure of Pyridine 4-Carbaldehyde Semicarbazone Schiff Base Ligand. Adv. J. Chem. Sec. A, 2020, 3, 534-541. DOI:10.33945/SAMI/AJCA.2020.4.14

[57] Balachandran, V.; Santhi, G.; Karpagam, V.; Lakshmi, A. Molecular structure, spectroscopic (FT-IR, FT-Raman), NBO and HOMO–LUMO analyses, computation of thermodynamic functions for various temperatures of 2, 6-dichloro-3-nitrobenzoic acid. Spectrochim. Acta Part A: Mol. and Biomol. Spect., 2013, 110, 130–140. https://doi.org/10.1016/j.saa.2013.03.021

[58] Srikanth, K. E.; Veeraiah, A.; Pooventhiran, T.; Thomas, R.; Solomon, K. A.; Raju, C. J. S.; Naveena, J.; Latha, L. Detailed molecular structure (XRD), conformational search, spectroscopic characterization (IR, Raman, UV, fluorescence), quantum mechanical properties and bioactivity prediction of a pyrrole analogue. Heliyon, 2020, 6, e04106-e04117. https://doi.org/10.1016/j.heliyon.2020.e04106

[59] Karpagakalyaani, G.; Daisy Magdaline, J.; Chithambarathanu, T.; Aruldhas, D.; Ronaldo Anuf, A. Spectroscopic (FT-IR, FT-Raman, NBO) Investigation and Molecular Docking study of a Herbicide compound Bifenox. Chem. Data Collect., 2020, 27, 100393. https://doi.org/10.1016/j.cdc.2020.100393

[60] Sundaram, M. S. S.; Karthick, S.; Sailaja, K.; Karkuzhali, R.; Gopu, G. Theoretical study on cyclophane amide molecular receptors and its complexation behavior with TCNQ. J. Photochem. Photobio, B: Biology, 2020, 203, 111735. https://doi.org/10.1016/j.jphotobiol.2019.111735

[6[1]] Merrick, J. P.; Moran, D.; Radom, L. An Evaluation of Harmonic Vibrational Frequency Scale Factors. J. Phys. Chem. A, 2007, 111, 11683-11700.

[62] Karabacak, M.; Bilgili, S.; Atac, A. Theoretical study on molecular structure and vibrational analysis included FT-IR, FT-Raman and UV techniques of 2,4,5-trimethylbenzoic acid (monomer and dimer structures). Spectrochim. Acta Part A: Mol. and Biomol. Spect., 2015, 134, 598-607.

[63] Silverstein, R. M.; Webster, F. X. Spectroscopic Identification of Organic Compound, sixth ed., John Willey & Sons, New York, 1998.

[64] Stuart B. H. Infrared Spectroscopy: Fundamentals and Applications, John Wiley & Sons, England, 2004.

[65] El‐Gammal, O. A.; Abu El‐Reash, G. M.; Bedier, R. A. Synthesis, spectroscopic, DFT, biological studies and molecular docking of oxovanadium (IV), copper (II) and iron (III) complexes of a new hydrazone derived from heterocyclic hydrazide. Appl Organometal. Chem., 2019, 33, e5141. https://doi.org/10.1002/aoc.5141

[66] Bharati, N.; Shailendra; Sharma, S.; Naqvi F.; Azam, A. New palladium(II) complexes of 5-nitrothiophene-2-carboxaldehyde thiosemicarbazones. synthesis, spectral studies and in vitro anti-amoebic activity. Bioinorg. Med. Chem., 2003, 11, 2923-2929. DOI: 10.1016/s0968-0896(03)00213-x

[67] Varsanyi, G. Vibrational Spectra of 700 Benzene Derivatives, vols. (I-II), Academic Kiego, Budapest, 1974.

[68] Dubis, A. T.; Grabowski, S. J.; Romanowska, D. B.; Misiaszek, T.; Leszczynski, J. Pyrrole-2-carboxylic Acid and Its Dimers:  Molecular Structures and Vibrational Spectrum. J. Phys. Chem. A, 2002, 106, 10613-10621.   https://doi.org/10.1021/jp0211786

[69] Li, Y.; Zhang, Y.; Niu, H.; Wang, C.; Qin, C.; Bai, X.; Wang, W. Schiff bases containing triphenylamine and pyrrole units: synthesis and electrochromic, acidochromic properties. New J. Chem., 2016, 40, 5245-5254. https://doi.org/10.1039/C6NJ00321D

[70] Tanak, H.; Ağar A. A.; Büyükgüngör, O. Experimental (XRD, FT-IR and UV–Vis) and theoretical modeling studies of Schiff base (E)-N’-((5-nitrothiophen-2-yl)methylene)-2-phenoxyaniline. Spectrochim. Acta Part A: Mol. and Biomol. Spect., 2014, 118, 672-682. http://dx.doi.org/10.1016/j.saa.2013.08.054

[71] Maryam, K.; Safar Ali, B.; Azar, G. Novel Schiff Bases of Pyrrole: Synthesis, Experimental and Theoretical Characterizations, Fluorescent Properties and Molecular Docking. Iran. J. Chem. Chem. Eng., 2018, 37 (6), 59-72.

[72] Azad, I.; Akhter, Y.; Khan, T.; Azad, M. I.; Chandra, S.; Singh, P.; Kumar, D.; Nasibullah, M. Synthesis, quantum chemical study, AIM simulation, in silico ADMET profile analysis, molecular docking and antioxidant activity assessment of aminofuran derivatives. J. Mol. Struct., 2020, 1203, 127285. https://doi.org/10.1016/j.molstruc.2019.127285

[73] Cuenú, F.; Londoño-Salazar, J.; Torres, J. E.; Abonia, R.; D'Vries, R. F. Synthesis, structural characterization and theoretical studies of a new Schiff base 4-(((3-(tert-Butyl)-(1-phenyl) pyrazol-5-yl) imino) methyl) phenol. J. M. Struct., 2018, 1152, 163-176. https://doi.org/10.1016/j.molstruc.2017.09.078

[74] Singh, R. N.; Rawat, P.; Verma, D.; Bharti, S. K. Experimental and DFT study on pyrrole tosylhydrazones. J. Mol. Struct., 2015, 1081, 543-554. https://doi.org/10.1016/j.molstruc.2014.10.046

[75] Anđelković, K.; Sladić, D. Complexes of iron(II), iron(III) and zinc(II) with condensation derivatives of 2-acetylpyridine and oxalic or malonic dihydrazide. Crystal structure of tris[(1-(2-pyridyl)ethylidene)hydrazine]iron(II) perchlorate. Transition Metal Chemistry, 2005, 30, 243–250. DOI:10.1007/s11243-004-3173-1

[76] Cuenú, F.; Restrepo-Acevedo, A.; Murillo, M. I.; Torres, J. E.; Moreno-Fuquen, R.; Abonia, R.; Kennedy, A. R.; Tenorio, J. C.; Lehmann, C. W. Synthesis, structural characterization, and theoretical studies of new pyrazole (E)-2-{[(5-(tert-butyl)-1H-pyrazol-3- yl)imino]methyl}phenol and (E)-2-{[(1-(4-bromophenyl)-3-(tert-butyl)-1H-pyrazol-5-yl] imino]methyl}phenol. J. Mol. Struct., 2019, 1184, 59–71.

[77] Gökce, H.; Öztürk, N.; Kazıcı, M.; Yörür Göreci, Ç.; Güneş, S. Structural, spectroscopic, electronic, nonlinear optical and thermodynamic properties of a synthesized Schiff base compound: A combined experimental and theoretical approach. J. Mol. Struct., 2017, 1136, 288-302. https://doi.org/10.1016/j.molstruc.2017.01.089

[78] Lien, E. J.; Guo, Z. R.; Li, R.; Su, C. T. Use of dipole moment as a parameter in drug-receptor interaction and quantitative structure-activity relationship studies. J. Pharm. Sci., 1982, 71, 641-655. DOI: 10.1002/jps.2600710611

[79] Pramodh, B.; Chethan Prathap, K. N.; Hema, M. K.; Warad, I.; Lokanath, N. K. Synthesis, structure, quantum computational and biological studies of novel thiophene derivatives. J. Mol. Struct., 2021, 1229, 129587.

[80] Snyder, S. H.; Merril, C. R. A relationship between the hallucinogenic activity of drugs and their electronic configuration. Proc. Natl. Acad. Sci. USA, 1965, 54, 258-266.

[81] Aihara, J. Reduced HOMO−LUMO Gap as an Index of Kinetic Stability for Polycyclic Aromatic Hydrocarbons. J. Phys. Chem. A, 1999, 103, 7487-7495.

[82] Li, J.; Cramer, C. J.; Truhlar, D. G. MIDI! basis set for silicon, bromine, and iodine. Theoret. Chem. Acc., 1998, 99 (3), 192-196. https://doi.org/10.1007/s002140050323

[83] Parr, R. G.; Yang, W. Density-Functional Theory of Atoms and Molecules. Oxford University Press, 1989.

[84] Pearson, R. G. The HSAB Principle - more quantitative aspects. Inorg. Chim. Acta, 1995, 240, 93-98. https://doi.org/10.1016/0020-1693(95)04648-8

[85] Pearson, R. G. Absolute electronegativity and hardness correlated with molecular orbital theory. Proceedings of the National Academy of Sciences of the United States of America, 1986, 83, 8440-8441. https://doi.org/10.1073/pnas.83.22.8440

[86] Pavitha, P.; Prashanth, J.; Ramu, G.; Ramesh, G.; Mamatha, K.; Venkatram, R. B. Synthesis, structural, spectroscopic, anti-cancer and molecular docking studies on novel 2-[(Anthracene-9-ylmethylene)amino]-2-methylpropane-1,3-diol using XRD, FTIR, NMR, UV-Vis spectra and DFT. J.  Mol. Struct., 2017, 1147, 406. DOI: 10.1016/j.molstruc.2017.06.095

[87] Domingo, L. R.; Aurell, M. J.; Perez, P.; Conteras, R. Quantitative Characterization of the Local Electrophilicity of Organic Molecules. Understanding the Regioselectivity on Diels−Alder Reactions. J. Phys. Chem. A., 2002, 106, 6871-6875. https://doi.org/10.1021/jp020715j

[88] Scrocco, E.; Tomasi, J. Electronic Molecular Structure, Reactivity and Intermolecular Forces: An Euristic Interpretation by Means of Electrostatic Molecular Potentials. Adv. Quant. Chem., 1978, 11, 115-193.

[89] Luque, F.J.; López, J.M.; Orozco, M. Electrostatic Interactions of a Solute with a Continuum. A Direct Utilization of AB initio Molecular Potentials for the Prevision of Solvent Effects. Theor. Chem. Acc., 2000, 103, 343-345. http://dx.doi.org/10.1007/s002149900013

[90] Shukla, R.; Bandopadhyay, P.; Sathe, M.; Chopra, D. Quantitative investigation on the intermolecular interactions present in 8-(4-ethoxyphenyl)-1,3-dimethyl-3,7-dihydro-1H-purine-2,6-dione with insight from interaction energies, energy framework, electrostatic potential map and fingerprint analysis. J. Chem. Sci., 2020, 132, 19-26.

[9[1]] Acharya, R.; Chacko, S.; Bose, P.; Lapenna, A.; Pattanayak, S. P. Structure Based Multitargeted Molecular Docking Analysis of Selected Furanocoumarins against Breast Cancer. Sci. Rep., 2019, 9, 15743-15756. https://doi.org/10.1038/s41598-019-52162-0

[92] Ahmed, M. J.; Dhumad, A. M.; Almashal, F. A.; Alshawi. J. M. Microwave-assisted synthesis, molecular docking and anti-HIV activities of some drug-like quinolone derivatives. Medi. Chem. Res., 2020, 29, 1067-1076. DOI: 10.1007/s00044-020-02546-z

[93] Thirumurugan, C.; Vadivel, P.; Lalitha, A.; Lakshmanan, S. Synthesis, characterization of novel quinoline-2-carboxamide based chalcone derivatives and their molecular docking, photochemical studies. Synthetic Communications, 2020, 50(6), 831-839. https://doi.org/10.1080/00397911.2020.1720737

[94] Shweta; Khan, E.; Tandon, P.; Maurya, R.; Kumar, P. A theoretical study on molecular structure, chemical reactivity and molecular docking studies on dalbergin and methyldalbergin. J. Mol. Struct., 2019, 1183, 100-106.

[95] Collins, D. M.; Conlon, N.T.; Kannan, S.; Verma, C.S.; Eli, L.D.; Lalani, A.S.; Crown, J. Preclinical Characteristics of the Irreversible Pan-HER Kinase Inhibitor Neratinib Compared with Lapatinib: Implications for the Treatment of HER2-Positive and HER2-Mutated Breast Cancer. Cancers, 2019, 11, 737. https://doi.org/10.3390/cancers11060737

[96] Yuan, Y.; Jin, W.; Nazir, Y.; Fercher, C.; Thomas Blaskovich, M.A.; Cooper, M.A.; Barnard, R.T.; Ziora, Z.M.Tyrosinase inhibitors as potential antibacterial agents. E. J. Med. Chem., 2020, 187, 111892. DOI: 10.1016/j.ejmech.2019.111892Y.

[97] Kang, S. M.; Heo, S. J.; Kim, K. N.; Lee, S. H.; Yang, H. M.; Kim, A. D.; Jeon, Y. J. Design, synthesis, molecular docking and biological evaluation of imides, pyridazines, imidazoles derived from itaconic anhydride for potential antioxidant and antimicrobial activities. Bioorg. Med. Chem., 2012, 20, 311-316.

[98] Nokinsee, D.; Shank, L.; Lee, V. S.; Nimmanpipug, P. Estimation of Inhibitory Effect against Tyrosinase Activity through Homology Modeling and Molecular Docking. Enzyme Research, 2015, 2015, 1-12. http://dx.doi.org/10.1155/2015/262364

[99] Cardoso, R.; Valente, R.; Souza da Costa, C.H.; da S. Gonçalves Vianez, J.L., Jr.; Santana da Costa, K.; de Molfetta, F.A.; Nahum Alves, C. Analysis of Kojic Acid Derivatives as Competitive Inhibitors of Tyrosinase: A Molecular Modeling Approach. Molecules, 2021, 26, 2875. https://doi.org/10.3390/molecules26102875 

[100] Gurunanjappa, P.; Kameshwar, V. H.; Kariyappa, A. K. Bioactive formylpyrazole Analogues: Synthesis, Antimicrobial, Antioxidant and Molecular docking studies. Asian Journal of Chemistry, 2017, 29 (7), 1549-1554.

[[1]01] Nayak, P.S.; Narayana, B.; Sarojini, B.K.; Sheik, S.; Shashidhara, K.S.; Chandrashekar, K.R. Design, synthesis, molecular docking and biological evaluation of imides, pyridazines, and imidazoles derived from itaconic anhydride for potential antioxidant and antimicrobial activities. Journal of Taibah University for Science, 2014, 10, 823-838.

http://dx.doi.org/10.1016/j.jtusci.2014.09.005

 [102] Murugavel, S.; Deepa, S.; Ravikumar, C.; Ranganathand, R.; Alagusundaram, P. Synthesis, structural, spectral and antibacterial activity of 3,3a,4,5-tetrahydro-2H-benzo[g]indazole fused carbothioamide derivatives as antibacterial agents. J. Mol. Struct., 2020, 1222, 128961. https://doi.org/10.1016/j.molstruc.2020.128961

Dr. Md Abdul Majed Patwary

Comilla University